# Single-phase deep learning in cortico-cortical networks

**Will Greedy**[*]
Bristol Computational Neuroscience Unit
Department of Computer Science, SCEEM
University of Bristol, United Kingdom
will.greedy@bristol.ac.uk

**Heng Wei Zhu**[*]
Bristol Computational Neuroscience Unit
School of Phys., Pharm. and Neuroscience
University of Bristol, United Kingdom
hengwei.zhu@bristol.ac.uk

**Joseph Pemberton**
Bristol Computational Neuroscience Unit
Department of Computer Science, SCEEM
University of Bristol, United Kingdom
joe.pemberton@bristol.ac.uk

**Jack Mellor**
School of Phys., Pharm. and Neuroscience
University of Bristol, United Kingdom
jack.mellor@bristol.ac.uk

**Rui Ponte Costa**
Bristol Computational Neuroscience Unit
Department of Computer Science, SCEEM
University of Bristol, United Kingdom
rui.costa@bristol.ac.uk

## Abstract

The error-backpropagation (backprop) algorithm remains the most common solution to the credit assignment problem in artificial neural networks. In neuroscience, it is unclear whether the brain could adopt a similar strategy to correctly modify its synapses. Recent models have attempted to bridge this gap while being consistent with a range of experimental observations. However, these models are either unable to effectively backpropagate error signals across multiple layers or require a multi-phase learning process, neither of which are reminiscent of learning in the brain. Here, we introduce a new model, Bursting Cortico-Cortical Networks (BurstCCN), which solves these issues by integrating known properties of cortical networks namely bursting activity, short-term plasticity (STP) and dendrite-targeting interneurons. BurstCCN relies on burst multiplexing via connection-type-specific STP to propagate backprop-like error signals within deep cortical networks. These error signals are encoded at distal dendrites and induce burst-dependent plasticity as a result of excitatory-inhibitory top-down inputs. First, we demonstrate that our model can effectively backpropagate errors through multiple layers using a single-phase learning process. Next, we show both empirically and analytically that learning in our model approximates backprop-derived gradients. Finally, we demonstrate that our model is capable of learning complex image classification tasks (MNIST and CIFAR-10). Overall, our results suggest that cortical features across sub-cellular, cellular, microcircuit and systems levels jointly underlie single-phase efficient deep learning in the brain.

---

[*]Equal contributions

36th Conference on Neural Information Processing Systems (NeurIPS 2022).

# 1  Introduction

For effective learning, synaptic modifications throughout the brain should result in improved behavioural function. This requires a process by which credit should be assigned to synapses given their contribution to behavioural output [1–3]. In multilayer networks, credit assignment is particularly challenging as the impact of changing a synaptic connection depends on its downstream brain areas. Classical local Hebbian plasticity rules, even when coupled with global neuromodulatory factors, are unable to communicate enough information for effective credit assignment through multiple layers of processing [3]. In machine learning, the error-backpropagation (backprop) algorithm is the most successful solution to the credit assignment problem. However, it relies on a number of biologically implausible assumptions to compute gradient information used for synaptic updates. Previous work has attempted to address these implausibilities but important issues remain open when mapping backprop to the neuronal physiology. Earlier attempts relied on using single-compartment neuron models [4, 5] but this poses a problem as single-compartment neurons are unable to simultaneously store the necessary inference and credit assignment signals. One solution is to model neurons with an apical dendritic compartment that separately stores credit information [5, 6], supported by the electrotonic separation of the soma and apical dendrites [7]. These distal credit signals can then be communicated to the soma through non-linear dendritic events that trigger bursting at the soma [8], thereby inducing long-term synaptic plasticity [9]. In particular, two recent approaches, Error-encoding Dendritic Networks (EDNs) [6] and Burstprop [10], have demonstrated how such multi-compartment neuron models can be used to construct networks capable of backprop-like credit assignment. EDNs encode credit signals at apical dendrites resulting from the mismatch between dendritic-targeting interneuron activity and downstream activity. Burstprop uses bursting, controlled by dendritic excitability, as a mechanism to communicate credit signals. However, these models still have major issues, such as the inability to effectively backpropagate error signals through many layers (EDNs) and the requirement for a multi-phase learning process (Burstprop).

Here, we propose a new model called the Bursting Cortico-Cortical Network (BurstCCN) as a solution to the credit assignment problem which addresses several outstanding issues of current biologically plausible backprop research. Our model builds upon prior multi-compartment neuron models [6, 10]: it encodes credit signals in distal dendritic compartments which trigger bursting activity at the soma to drive backprop-like synaptic updates. We demonstrate that combining well-established properties of cortical neurons such as bursting activity, short-term plasticity (STP) and dendrite-targeting interneurons provides a biologically plausible mechanism for performing credit assignment. In contrast to previous models, BurstCCN is highly effective at backpropagating credit signals in multi-layer architectures while only requiring a single-phase learning process. We implement multiple versions of the BurstCCN at different levels of abstraction in order to demonstrate some of its key properties and to empirically confirm our theoretically motivated claims.

First, we use a spike-based implementation of the BurstCCN to demonstrate its ability to learn without the need for multiple phases. We further show the importance of this single-phase learning by training a continuous-time rate-based version of the BurstCCN on a continuous-time non-linear regression task. Next, we show both empirically and analytically that our model's dynamics result in learning that approximately follows backprop-derived gradients. Finally, we use a simplified discrete-time BurstCCN implementation to demonstrate that the model achieves good performance on non-trivial image classification tasks (MNIST and CIFAR-10), even in the presence of random feedback synaptic weights.

# 2  Bursting Cortico-Cortical Networks

## 2.1  Burst Ensemble Multiplexing

Burst Ensemble Multiplexing (BEM) [11] refers to the idea that ensembles of cortical neurons are capable of simultaneously representing multiple distinct signals within the patterns of their spiking activity. Typically, pyramidal cells receive top-down and bottom-up signals into their apical and basal dendrites, respectively. Bottom-up basal inputs affect the rate of spiking and top-down apical inputs convert these somatically induced spikes into high-frequency bursts. Postsynaptic populations can then use STP to decode these distinct signals from the overall spiking activity.

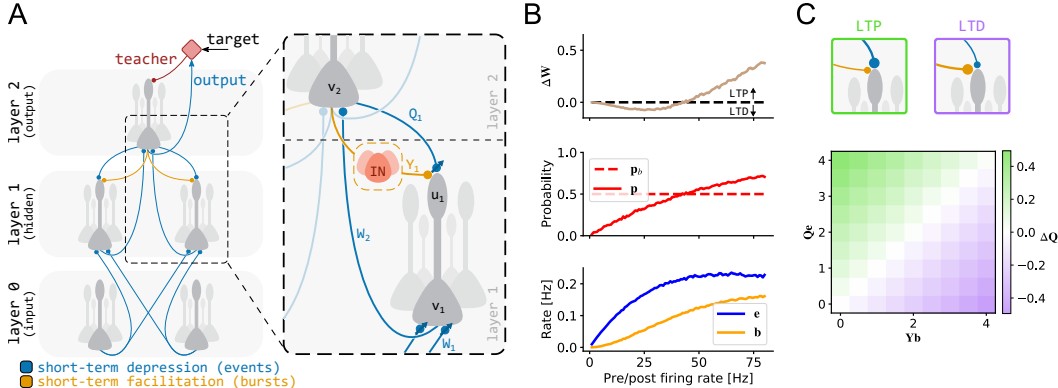

Figure 1: **Bursting cortico-cortical networks (BurstCCN) for credit assignment through bursting activity.** (**A**) Network schematic consisting of neuron ensembles and connection-type-specific STP. Events from the input are propagated forward through short-term depressing (STD) connections, $\mathbf{W}$. Output event rates are compared to a target value which generates a teaching signal that is presented to the output layer apical dendrites. This acts as an error signal and appears as a deflection in the dendritic potential from its resting potential which causes changes to bursting activity from its baseline. The error-carrying bursting signals are propagated back through short-term facilitating connections, $\mathbf{Y}$, which we interpret as being communicated by populations of dendrite-targeting interneurons. Events are also propagated backwards via STD connections, $\mathbf{Q}$, to provide a means of cancelling baseline bursting activity. The difference in activity from these two feedback connections results in changes to dendritic excitability that lead to burst-dependent synaptic plasticity. (**B**) Burst-dependent plasticity rule. Simple setup of a single connection between a pre- and post-synaptic cell that are both modelled with Poisson spike trains with equal rates. As the firing rates increase, (top) plasticity of the synaptic weight switches from long-term depression (LTD) to long-term potentiation (LTP) (middle) when the burst probability increases above the baseline value. (bottom) The magnitude of the weight change is scaled by the event rate. (**C**) Homeostatic plasticity rule for $\mathbf{Q}$ weights. The difference between the signals through $\mathbf{Q}$ and $\mathbf{Y}$ dictates the direction and magnitude of synaptic plasticity.

The BurstCCN uses the concept of BEM in a similar way to Burstprop [10] in which ensembles of cells encode both feedforward inference signals and feedback error signals. The model encodes these signals as the rates of *events* and *bursts*, respectively, across the ensembles. Here, the specific definition of a burst is a collection of spikes with interspike intervals less than 16ms and an event is either a burst or a single isolated spike (i.e. a spike not followed or preceded by another within 16ms). The burst probability of an ensemble is defined as the probability that an event at a given time is a burst and is computed as a ratio of the event rate (e) and burst rate (b): $\mathbf{p} = \mathbf{b}/\mathbf{e}$.

## 2.2 Rate-based BurstCCN

In our discrete-time implementation of the rate-based BurstCCN, example input-output pairs are processed independently in discrete timesteps. For each example, the event rates of the input layer, $\mathbf{e}_0$, encode the input stimulus. The model then computes each subsequent layer's activities, equivalent to that of a standard feedforward artificial neural network (Fig. 1A). Specifically, somatic potentials are computed by integrating basal input as $\mathbf{v}_l = \mathbf{W}_l \mathbf{e}_{l-1}$ where $\mathbf{W}_l$ are short-term depressing (STD) feedforward weights from layer $l - 1$ to layer $l$. The STD nature of these weights ensures that only event rate information propagates forwards. Each layer's event rates are then computed by applying a non-linear activation function, $f$, to the somatic potentials, $\mathbf{e}_l = f(\mathbf{v}_l)$. These linear and nonlinear operations are repeated for each layer in the network to ultimately obtain the output layer event rates, $\mathbf{e}_L$, where $L$ denotes the total number of layers.

The desired target output of the network, $\mathbf{e}_{target}$, is compared to the output layer event rates to produce a signed error, $\mathbf{e}_{target} - \mathbf{e}_L$, which is used as a teaching signal. This error information is then propagated backwards through each layer in the network by altering the apical dendritic compartment potential and, as a result, the burst probability of each pyramidal ensemble. At the output layer, the burst probability is computed directly as $\mathbf{p}_L = \mathbf{p}_L^b + \mathbf{p}_L^b \odot (\mathbf{e}_{target} - \mathbf{e}_L) \odot h(\mathbf{e}_L)$ where $\odot$

denotes the element-wise product, $\mathbf{p}_L^b$ represents the baseline burst probability in the absence of any teaching signal and $h(\mathbf{e}_l) = f'(\mathbf{v}_l) \odot \mathbf{e}_l^{-1}$. These burst probabilities are used at the output layer ($l = L$) to compute the burst rates as $\mathbf{b}_l = \mathbf{e}_l \odot \mathbf{p}_l$ which are decoded and sent backwards to layer $l - 1$ apical dendrites by a set of short-term facilitating (STF) feedback weights, $\mathbf{Y}_{l-1}$. The STF feedback weights and STD feedforward weights are similarly used in Burstprop. However, the BurstCCN additionally includes a novel set of apical dendrite-targeting STD feedback weights, $\mathbf{Q}_{l-1}$, which send event rates backwards. We interpret the STF feedback connections as being provided via a type of dendrite-targeting interneuron and STD feedback as direct connections in line with recent experimental studies [12–17]. The signals through both sets of feedback weights lead to the apical potentials in the previous layer, $\mathbf{u}_{l-1} = \mathbf{Q}_{l-1}\mathbf{e}_l - \mathbf{Y}_{l-1}\mathbf{b}_l$. These determine the layer's burst probabilities which are computed as $\mathbf{p}_{l-1} = \bar{\sigma}(\mathbf{u}_{l-1} \odot h(\mathbf{e}_{l-1}))$ where $\bar{\sigma}$ denotes the sigmoid function, $\sigma$, with scaling and offset parameters, $\bar{\sigma}(x) = \sigma(\alpha x + \beta)$ ([10]; see SM, Section B). The same process is repeated backwards for each layer until the input layer to obtain their dendritic potentials and burst probabilities. Note that for all experiments, we set $\alpha = 4$ (and $\beta = 0$) to prevent this function from implicitly scaling down the errors propagating backwards through each layer by a factor of 4 (since $\frac{d\sigma}{dx} \approx \frac{1}{4}$ around $x = 0$).

After the error information has been propagated backwards, feedforward synaptic weight changes are computed using a burst-dependent synaptic plasticity rule:

$$\Delta\mathbf{W}_l = \eta_l^{(\mathbf{W})} \left((\mathbf{p}_l - \mathbf{p}_l^b) \odot \mathbf{e}_l\right)\mathbf{e}_{l-1}^T \tag{1}$$

where $\eta^{(\mathbf{W})}$ is a learning rate and $\cdot^T$ is the transpose operation. Importantly, the learning rule depends on the change in burst probability from the predefined layer-wise baseline burst probability, $\mathbf{p}_l^b = p_l^b(1, \ldots, 1)^T$, which represents the signed error signal required for backprop-like learning. Consequently, when we make both pre- and postsynaptic cells fire following Poisson statistics we obtain long-term depression and long-term potentiation for low and high firing rates, respectively (Fig. 1B). This is in line with a large number of experimental studies of cortical synapses [18, 19]. It can be shown that the updates produced by this learning rule approximate those obtained by the backpropagation algorithm in the weak-feedback case (see Section 3.3.1 and SM, Section B).

In the absence of a teaching signal, it is important for pyramidal ensembles to produce a baseline level of bursting such that no weight changes occur (cf. Eq. 1). This holds true for the output layer as there are no other inputs onto the apical dendrites. However, for the hidden layers the event rate signals through $\mathbf{Q}$ and the burst rate signals through $\mathbf{Y}$ need to exactly cancel each other out such that the apical dendritic potentials are at rest (i.e. $\mathbf{u} = 0$). For any $\mathbf{Y}$ weights, there is always an optimal set of $\mathbf{Q}$ weights that will produce this exact cancellation regardless of the event rates propagating through the network. Specifically, they must be set as $\mathbf{Q}_l = p_l^b \mathbf{Y}_l$ which we refer to as the weights being in a *Q-Y symmetric* state. However, it is not biologically plausible for the $\mathbf{Q}$ synapses to have direct knowledge of $\mathbf{Y}$. Therefore, inspired by earlier work [6, 20], we use a learning rule for $\mathbf{Q}$ to provide this cancellation:

$$\Delta\mathbf{Q}_l = -\eta_l^{(\mathbf{Q})} \mathbf{u}_l \mathbf{e}_{l+1}^T \tag{2}$$

which explicitly aims to silence the apical potentials (Fig. 1C). In the absence of a teaching signal at the output layer, all $\mathbf{Q}$ weights will eventually converge to their optimal values and achieve a symmetric state under reasonable assumptions (see SM, Section B.2). Note that we could similarly have added this learning rule on the $\mathbf{Y}$ feedback weights to cancel the activity through the $\mathbf{Q}$ weights, which produces similar results (Fig. S1).

When teaching signals are applied at the output layer, it is important to note that only the bursting activity propagated through the $\mathbf{Y}$ connections changes because the event rates through $\mathbf{Q}$ are unaffected by the dendritic activity. This enables single-phase learning as the symmetry in the two feedback connection types ($\mathbf{Q}$ and $\mathbf{Y}$) can be exploited to directly compare *without teacher* signals (i.e. at baseline) to *with teacher* signals.

Details of the continuous time implementation can be found in the Supplementary Materials.

## 2.3 Spiking BurstCCN

For our spiking implementation of the BurstCCN, we adapted the burst-dependent synaptic plasticity rule in Equation 1 (see SM, Eq. 12). Unlike the two rate-based implementations, the spiking BurstCCN more accurately models the internal neuron spiking dynamics instead of abstracting these

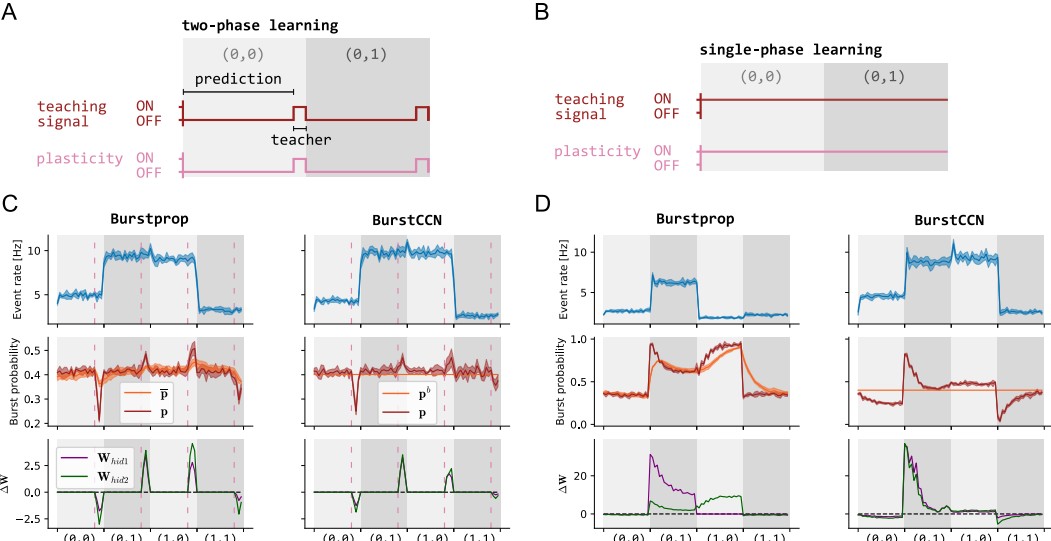

Figure 2: **Spiking BurstCCN does not require multi-phase learning to solve the XOR classification task.** Schematic of the (A) two-phase and (B) single-phase learning settings. (**A**) For each input during two-phase learning, networks are given a 7.2s prediction period during which teaching signals and plasticity are turned OFF, followed by a 0.8s learning period where both teaching signals and plasticity are turned back ON. (**B**) During single-phase learning, both the teaching signals and plasticity remain ON throughout training. (**C, D**) Top: event rate ($e$) of the output layer. Middle: burst probability ($p$) for the output layer and the baseline or moving average of the burst probability ($p^b$ or $\bar{p}$) for BurstCCN and Burstprop, respectively. Bottom: the resulting weight updates for connections from hidden layer neurons. Model results represent mean $\pm$ standard error (n = 5).

details away and only considering the ensemble-level behaviour. Neurons are modelled with two compartments corresponding to the soma and apical dendrites and spikes are generated when a somatic threshold potential is met (see SM, Section A.2 for more details).

## 2.4    Related work

As previously mentioned, BurstCCN takes inspiration from two prior models: EDNs [6] and Burstprop [10]. Similar to these models, the BurstCCN uses a separate apical dendritic compartment to represent an error signal. To silence this apical compartment and maintain correct error signals, the EDN uses a homeostatic plasticity rule from local interneurons to cancel the signals received from a separate feedback pathway. In the BurstCCN, we use the same principle by adapting this plasticity rule for learning of the novel **Q** weights. Unlike the EDN, we use a similar idea to Burstprop in which error signals are encoded as bursts in the neural activity and decoded by STP dynamics.

Within each layer, Burstprop includes a set of recurrent connections onto the apical compartments which aim to maintain the dendritic potential in the linear regime of the feedback non-linearity. Updating the weights of these connections requires separate learning phases and it is unclear how the plasticity rule can be justified. In contrast, the BurstCCN does not require these connections. Instead, the novel set of STD feedback connections (**Q**) onto the apical dendrites provide a mechanism for single-phase learning and perform a similar role of linearising the feedback. Additionally, burst-dependent plasticity in our model relies on a constant baseline burst probability instead of using a moving average of the burst probability (see SM, Section A.2 for more information).

## 3    Results

### 3.1    BurstCCN can learn with a single learning phase

A key motivation for developing the BurstCCN was to design a model capable of learning without the need for separate learning phases, while being consistent with a range of cortical features across

multiple levels. To demonstrate that our model can perform single-phase learning, we trained the spiking version of our model on the XOR classification task and contrasted it with Burstprop, which requires a two-phase learning process (Fig. 2). In both single- and two-phase learning regimes, the input stimulus is presented for a total of 8s before the next example is shown. The two-phase learning regime has an initial prediction phase, lasting 7.2s for each input presentation, where plasticity is switched off throughout the network and the output neurons do not receive any teaching signals (Fig. 2A). This is followed by a teacher phase for the remaining 0.8s where plasticity is restored and teaching signals are delivered at the output. The single-phase regime removes the initial prediction phase and extends the teacher phase to the full duration of the input stimulus (Fig. 2B).

Our results show that both models were capable of successfully learning the task in the two-phase regime as indicated by the high output event rates in response to the $(0, 1)$ and $(1, 0)$ inputs and low event rates for the $(0, 0)$ and $(1, 1)$ inputs (Fig. 2C). However, when training in the single-phase regime, only BurstCCN was able to learn the task (Fig. 2D). The inability of Burstprop to learn the task can be explained by comparing the moving average of the burst probability ($\overline{\mathbf{p}}$) with the actual burst probability ($\mathbf{p}$) which determines the sign of synaptic weight updates (Fig. 2D). Burstprop failed to learn in the single-phase learning setup due to the teaching signal remaining on and preventing $\overline{\mathbf{p}}$ from being able to provide a stable representation of the without-teacher burst probability.

### 3.2 BurstCCN can learn with dynamic input-output

Typically, studies that have attempted to solve the credit assignment problem with biologically plausible implementations of backprop make an implicit assumption that during learning there is a period where the continuous-time input stream is fixed [6, 10]. This is required in most cases to allow the network to stabilise its activities before learning can take place. With single-phase learning, we can relax this assumption to enable learning in conditions where the inputs and their corresponding teaching signals are dynamically changing over time. We assessed this ability by training the continuous-time BurstCCN (see SM, Section A.1) on an online non-linear regression task (Fig. 3). This task consisted of three sinusoidal inputs, $x_i(t) = sin(\alpha_i t + \beta_i)$, with random frequencies $\alpha_i \sim U(0, \frac{\pi}{2})$ and phase offsets $\beta_i \sim U(0, 2\pi)$ (Fig. 3A). The network had a single output unit for which a non-trivial target was obtained by passing the same inputs to a 3-25-1 artificial neural network (ANN). This approximates a setting in which a given cortical area learns to regress its input onto the activity of another cortical area. The ANN weights were randomly initialised with $w_{ij}^1 \sim \mathcal{U}(-\sqrt{3}, \sqrt{3})$ for the first layer and $w_{ij}^2 \sim \mathcal{U}(-0.6, 0.6)$ for the second layer. Despite the BurstCCN initially producing outputs that were significantly different to the target (Fig. 3C), the results show that over training it learned to produce output patterns that closely matched the non-linear and dynamic target (Fig. 3B,D). This highlights that the BurstCCN is capable of adequately backpropagating useful error signals when both inputs and teaching signals are constantly changing.

### 3.3 Feedback plasticity rule facilitates alignment to backprop updates

Next, we wanted to understand how well our model approximates backprop. As stated above, the purpose of the learning rule for the feedback STD $\mathbf{Q}$ connections (Eq. 2) is to silence the apical compartments in every ensemble by cancelling activity through the feedback STF $\mathbf{Y}$ connections. When a teaching signal is applied, this becomes important for computing the correct local error signal that is used for learning and backpropagated to previous layers. Here, we show both analytically and empirically using the discrete version of the model how the computed errors relate to backprop.

#### 3.3.1 BurstCCN with weak feedback approximates backpropagation algorithm

Under some small assumptions, we analytically show that the feedback pathway of BurstCCN is approximately communicating the same error gradients that are computed by backprop. Specifically, we assume that the feedback weights are optimally aligned (i.e. $\mathbf{Q}_l = p_l^b \mathbf{Y}_l$) and focus on the change in burst rate, $\delta\mathbf{b}_l := (\mathbf{p}_l - \mathbf{p}_l^b) \odot \mathbf{e}_l$. If we let $E^{\text{task}} = ||\mathbf{e}_L - \mathbf{e}_{target}||^2$ define the task error then, by construction, the change in burst rate at the output layer is equivalent to the negative error gradient, $\delta\mathbf{b}_L = -\frac{\partial E^{\text{task}}}{\partial \mathbf{v}_L}$. For the hidden layers, we derive the following iterative relationship (see SM, Section B):

$$\delta\mathbf{b}_l = f'(\mathbf{v}_l) \odot (-\mathbf{Y}_l)\delta\mathbf{b}_{l+1} + \mathcal{O}(\mathbf{u}_l^3). \tag{3}$$

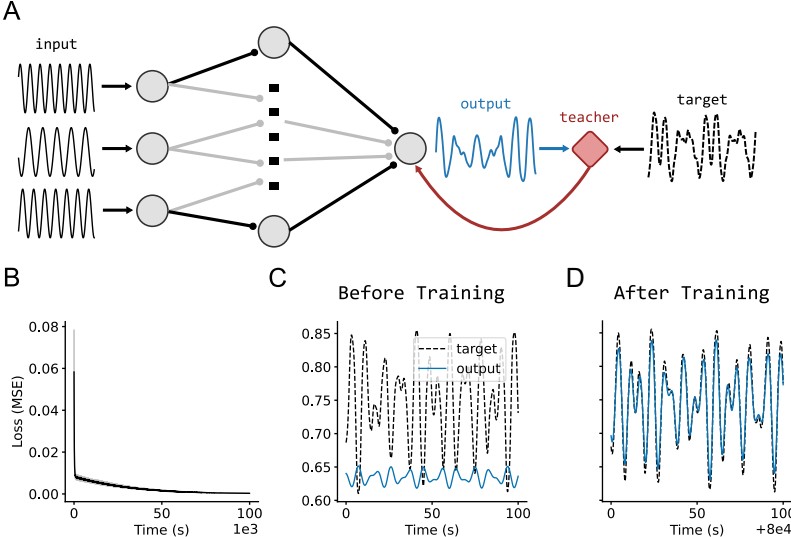

Figure 3: **BurstCCN can learn a dynamic non-linear regression task.** (**A**) Schematic of the task. Three sinusoidal waves with random frequencies are given as inputs. The task is to learn to match the target pattern which is obtained by passing the same inputs through a fixed, randomly initialised ANN. (**B**) Learning curve for the (continuous-time) BurstCCN. (**C, D**) Example output traces for (C) before and (D) after training. Model results represent mean $\pm$ standard error (n = 5).

This approximates the same relationship present in backprop up to a third-order[1] term with respect to the apical potentials $\mathbf{u}_l$ if the feedback weights are set to be symmetric with the feedforward weights (i.e. $\mathbf{Y}_l = -\mathbf{W}_{l+1}^T$). We refer to this as the *W-Y symmetric* state. The link between weight updates from simply performing gradient descent with backprop and the BurstCCN can be seen clearly:

$$\Delta \mathbf{W}_l^{\text{BurstCNN}} = \eta_l^{(\mathbf{W})} \, \delta \mathbf{b}_l \, \mathbf{e}_{l-1}^T \tag{4}$$

$$\Delta \mathbf{W}_l^{\text{backprop}} = -\eta_l^{(\mathbf{W})} \frac{\partial E^{\text{task}}}{\partial \mathbf{v}_l} \, \mathbf{e}_{l-1}^T \tag{5}$$

It remains to be shown that the apical potentials, $\mathbf{u}_l$, of every layer are indeed appropriately small (so that the approximation error, $\|\mathbf{u}_l^3\|$, is small). Under the assumption $\mathbf{u}_{l+1}$ is small, we can derive the recursive relationship $\mathbf{u}_l \approx f'(\mathbf{v}_{l+1}) \odot (-\mathbf{Y}_l)\mathbf{u}_{l+1}$ (see SM, Section B). We show that if $f'$ is bounded (as is the case for sigmoid and many activation functions) and the weights $\mathbf{Y}_l$ are reasonably small then $\|\mathbf{u}_l^3\| \leq \|\mathbf{u}_{l+1}^3\|$. This means that if the error gradient at the output layer, $\frac{\partial E^{\text{task}}}{\partial \mathbf{e}_N}$, is small then, by induction, $\|\mathbf{u}_l^3\|$ is small for every layer and $\Delta \mathbf{W}_l^{\text{BurstCNN}} \approx \Delta \mathbf{W}_l^{\text{backprop}}$.

### 3.3.2 Learning Q feedback connections better approximates backprop-derived gradients

We empirically evaluated our feedback plasticity rule by updating *only* the $\mathbf{Q}$ weights of a randomly initialised 5-layer discrete-time BurstCCN with all other weight types ($\mathbf{W}$ and $\mathbf{Y}$) fixed. We used multiple initialisations and training regimes to understand how the plasticity rule behaves in different scenarios. The network was either initialised in the $\mathbf{W}$-$\mathbf{Y}$ symmetric state or with random feedback weights (where $\mathbf{Y}_l \neq -\mathbf{W}_{l+1}^T$). We computed the angle between the update that would have been made by the feedforward plasticity rule (Eq. 1) and either backprop or feedback alignment [4] for the symmetric and random configurations, respectively. We examined both cases: in the theoretically ideal case for learning $\mathbf{Q}$ where no teaching signal is present (Fig. 4A-D) and with a teaching signal at the output layer (Fig. 4E-H).

In all cases, as the alignment between the $\mathbf{Q}$ and $\mathbf{Y}$ connections improved (Fig. 4A,E), the apical potential decreased (Fig. 4B,F) and this resulted in updates that more closely aligned to back-prop (Fig. 4C,G) and feedback alignment (Fig. 4D,H). In the absence of a teaching signal, this

---

[1] Here we use abuse of notation $\mathbf{u}_l^3 = \left(u_{l,1}^3, u_{l,2}^3, \dots\right)^T$ to represent the element-wise cubic of $\mathbf{u}_l$

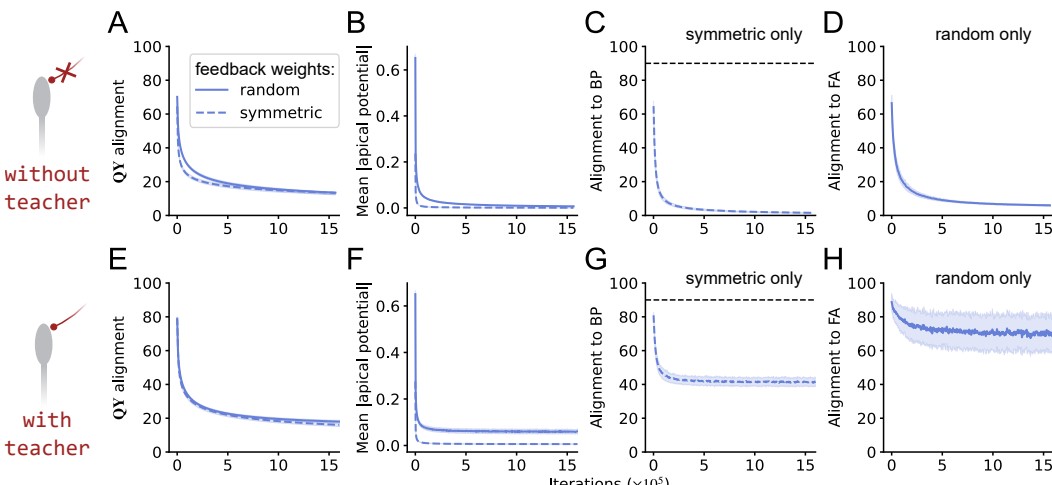

Figure 4: **Feedback learning rule enables a close alignment with backprop and feedback alignment.** The network is a randomly initialised 5-layer discrete-time BurstCCN with random (solid line) or symmetric (dashed line), fixed $\mathbf{W}$ and $\mathbf{Y}$ weights. The $\mathbf{Q}$ weights are updated in the presence of (**A-D**) no teaching signal or (**E-H**) a teaching signal. (A,E) Alignment between $\mathbf{Q}$ and $\mathbf{Y}$ weights, (B,F) the mean absolute value of the apical potentials, (C,G) the alignment to backprop (BP) and (D,H) feedback alignment (FA) as the $\mathbf{Q}$ weights learn to silence apical dendrite potential. Updates below 90° marked by the black dashed line are considered useful as they still follow the direction of backprop on average. Model results represent mean $\pm$ standard error (n = 5).

alignment angle to both backprop and feedback alignment eventually became very small which supports our analytical results that show our model approximates these methods (Fig. 4C-D). Despite producing less aligned feedforward updates in the presence of a teaching signal, the updates computed were still informative since they were consistently well below 90° of the direction of steepest descent (Fig. 4G).

### 3.4 BurstCCN learns image classification tasks with multiple hidden layers

#### 3.4.1 MNIST

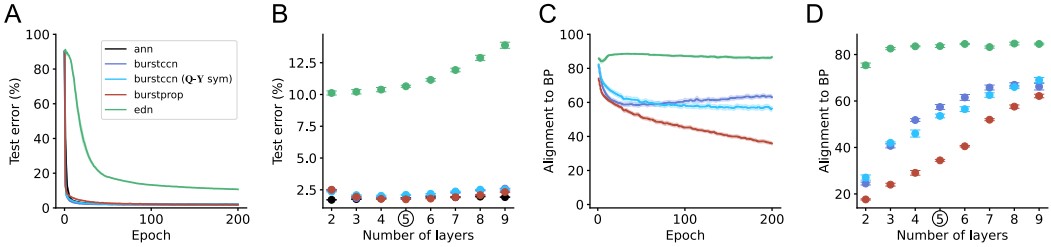

Figure 5: **BurstCCN learns to classify handwritten digits (MNIST) with deep networks.** (**A**) Learning curve of 5-layer ANN (black), BurstCCN (blue), BurstCCN ($\eta^{(\mathbf{Q})} = 0$) (light blue), Burstprop (red) and EDN (green). (**B**) Test error with different numbers of hidden layers for all models. (**C**) Alignment to backprop (BP) over time for all 5-layer models. (**D**) Alignment to backprop with different numbers of hidden layers for all models. The black circle indicates that the hyperparameters for each model were optimised for 5-layer networks. Model results represent mean $\pm$ standard error (n = 5).

Next, to test whether our model can indeed perform backprop-like deep learning, we trained a number of (discrete-time) BurstCCN architectures on the MNIST handwritten digit classification task [21]. We compared the BurstCCN with Burstprop [10] and EDNs [6] using similar architectures (see SM, Section C.3.3). We focused on the more biologically plausible case of using random fixed feedback

weights (i.e. feedback alignment [4]; see Fig. S2 for symmetric feedback weight case) with the remaining connection types of the different models updated using their respective plasticity rules. We also tested the BurstCCN in its idealised case where the feedback STD weights ($\mathbf{Q}$) were fixed in the $\mathbf{Q}$-$\mathbf{Y}$ symmetric state (see Section 2.2). We denote this model as "BurstCCN ($\mathbf{Q}$-$\mathbf{Y}$ sym)".

Using 5-layer networks, the BurstCCN obtained a test error of 1.84±0.01%, comparable to that of Burstprop with 1.75±0.01% and significantly outperforming the EDN with 10.65±0.09% (Fig. 5A). As the network depth was increased, both BurstCCN and Burstprop retained high performances but the EDN showed a substantial decay in performance with deeper networks (Fig. 5B). In an idealised case for the EDN, the disparity in performance and the effect of depth is less evident (Fig. S3). We then compared the alignment between the models and backprop. For the 5-layer networks, Burstprop's updates were most closely aligned to backprop, followed by the two BurstCCN models which all vastly outperformed the EDN (Fig. 5C). As expected, the BurstCCN with $\mathbf{Q}$-$\mathbf{Y}$ symmetry could better propagate error signals. By increasing the network depth, we demonstrate that it was more difficult to produce updates that were closely aligned to backprop. However, we show that the BurstCCN was still capable of backpropagating useful error signals in relatively deep networks (Fig. 5D).

### 3.4.2 CIFAR-10

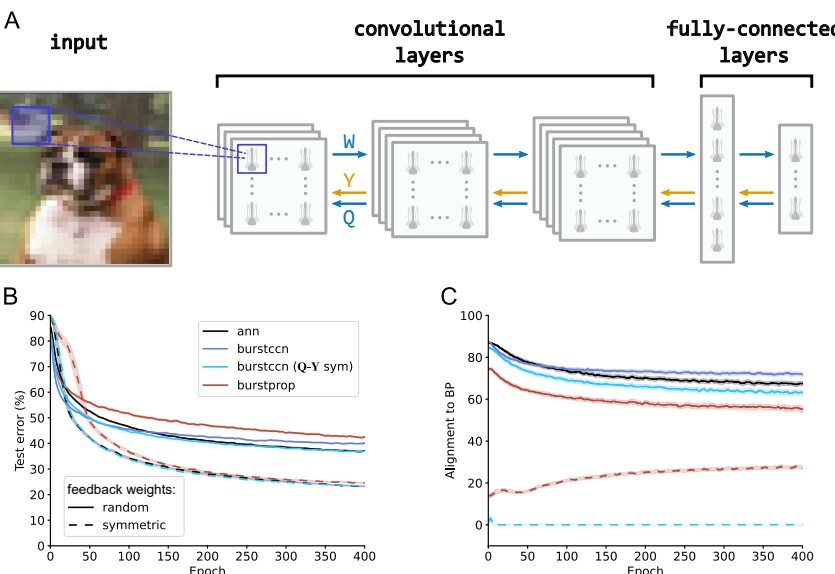

Figure 6: **BurstCCN with convolutional layers learns to solve natural image classification task (CIFAR-10).** (**A**) Schematic of BurstCCN architecture consisting of an input layer, three convolutional layers, a fully-connected hidden layer and output layer. For the BurstCCN, each layer was connected with a set of feedforward weights, $\mathbf{W}$, and feedback weights, $\mathbf{Y}$ and $\mathbf{Q}$. (**B**) Learning curve and (**C**) alignment to backprop of the different models with random (solid lines) and symmetric (dashed lines) feedback weight regimes. Model results represent mean ± standard error (n = 5).

Finally, we wanted to investigate the capabilities of the BurstCCN on more challenging tasks that are commonly tested in deep learning. We constructed a deep network consisting of three convolutional layers followed by a fully-connected hidden layer and output layer (Fig. 6A). We trained ANN, BurstCCN and Burstprop models using this network architecture on the CIFAR-10 image classification task [22]. BurstCCN ($\mathbf{Q}$-$\mathbf{Y}$ sym) was trained in the $\mathbf{Q}$-$\mathbf{Y}$ symmetric regime whereas BurstCCN was initialised in this state and $\mathbf{Q}$ weights were then updated using the corresponding plasticity rule. All model types were tested with two feedback weight regimes: $\mathbf{W}$-$\mathbf{Y}$ symmetric and random fixed $\mathbf{Y}$ feedback weights (i.e. feedback alignment).

After training in the random feedback weight regime, we observed a test error of 38.99±0.18% for BurstCCN, similar to performances achieved by an ANN (36.30±0.16%) and Burstprop (41.32±0.14%) (Fig. 6B). For the $\mathbf{W}$-$\mathbf{Y}$ symmetric regime which most resembles backprop, BurstCCN (22.92±0.03%) performed significantly better than all random feedback setups and, once

again, obtained a similar error to the symmetric ANN (22.62±0.10%) and Burstprop (24.15±0.17%) models. In the symmetric setups, there was a large improvement in the alignment angles to backprop compared to the random feedback setup (Fig. 6C). This suggests that they were backpropagating errors more effectively which likely explains the increase in performance. However, as seen within the random feedback setups, an improvement in this alignment does not guarantee an improvement to performance. This is because each model will traverse a different learning trajectory and converge to a different local minimum but the alignment angle remains a good indicator of expected performance.

## 4    Conclusions and discussion

We have introduced a new model capable of backprop-like credit assignment by integrating known properties of cortical networks. We have shown that by combining specific biological mechanisms such as bursting, STP and dendrite-targeting inhibition it is possible to construct a model that learns effectively in a continuous setting that is reminiscent of learning in the brain. Moreover, we have demonstrated that such a model can learn complex image classification tasks with deep networks.

Our model proposes specific STP dynamics on the feedforward and feedback connections. It requires STD on cortico-cortical projections onto pyramidal cells in line with experimental evidence [12–16]. In addition, it suggests a key role for dendrite-targeting interneurons such as SST-positive Martinotti cells in the feedback pathway. There is evidence that these interneurons receive STF top-down connections whereas top-down projections onto pyramidal cells exhibit STD dynamics as required by our model [12–17]. In future work, it would be interesting to model the specific neuron types for each connection to satisfy Dale's law and further increase biological plausibility.

A prediction from our model is that manipulations of interneurons with STF connections would lead to disruptions in burst decoding from the layer (brain area) above thereby obstructing learning in the brain area below. Additionally, as error signals alter the level of bursting in the network, the model predicts that the variance in bursting activity and the distal dendritic potentials would correlate with the severity of errors made by the network during learning.

Although our model captures a wide range of biological features, some biological implausibilities remain. Currently, we use feedback alignment to provide a solution to the *weight transport problem* [23] but this has a substantial impact on performance, particularly in more challenging tasks. Therefore, it would be important to explore some of the recently introduced plausible feedback learning rules [24–26] which could be used in conjunction with our proposed learning rules to outperform feedback alignment [4].

Overall, our work provides a novel solution to the credit assignment problem and suggests that a range of cortical features from sub-cellular to the systems level jointly underlie single-phase, efficient deep learning in the brain.

## Acknowledgments and Disclosure of Funding

The authors would like to thank Alexandre Payeur, Jordan Guerguiev, Blake Richards, Richard Naud, Kevin Nejad, Jesper Sjostrom, Paul Anastasiades, Joao Sacramento, Adil Khan and Jasper Poort for useful discussions. This work made use of the supercomputer BluePebble. We would also like to thank Callum Wright and the rest of the High Performance Computing team at the University of Bristol for constant and quick help with BluePebble. This work has been supported by two EPSRC Doctoral Training Partnership PhD studentships to Will Greedy and Joseph Pemberton and a Wellcome Trust Neural Dynamics PhD studentship to Heng Wei Zhu.

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
