## Supplementary Material

## A   Model details

### A.1   Continuous-time BurstCCN

In contrast to the discrete-time implementation, the continuous-time BurstCCN does not process each input example independently from one another at discretised timesteps. The network dynamics instead evolve through a continuous-time simulation incrementing with timesteps of $\Delta t = 0.1s$, where there is a memory of the prior network state at each timestep and no parallel processing of mini-batches. The signals into the input layer are now given by a time-varying function, $\mathbf{x}(t)$, along with target signals to the output layer, $\mathbf{y}(t)$. The input layer event rate is set instantaneously as $\mathbf{e}_0(t) = \mathbf{x}(t)$. All somatic potentials and other layer event rates evolve with:

$$\frac{d\mathbf{v}_l(t)}{dt} = \frac{1}{\tau_v}\Big(-\mathbf{v}_l(t) + \mathbf{W}_l(t)\mathbf{e}_{l-1}(t)\Big) \tag{6}$$

$$\mathbf{e}_l(t) = f(\mathbf{v}_l(t)) \tag{7}$$

where $\tau_v$ is the membrane leak time constant of the soma. The output layer burst probabilities are also set instantaneously as $\mathbf{p}_L(t) = \mathbf{p}_L^b + \mathbf{p}_L^b \odot (\mathbf{y}(t) - \mathbf{e}_L(t)) \odot h(\mathbf{e}_l(t))$. The hidden layer dendritic potentials, burst probabilities and burst rates evolve with:

$$\frac{d\mathbf{u}_l(t)}{dt} = \frac{1}{\tau_u}\Big(-\mathbf{u}_l(t) + h(\mathbf{e}_l(t)) \odot \Big[\mathbf{Q}_l(t)\mathbf{e}_{l+1}(t) - \mathbf{Y}_l(t)\mathbf{b}_{l+1}(t)\Big]\Big) \tag{8}$$

$$\mathbf{p}_l(t) = \sigma(\mathbf{u}_l(t)) \tag{9}$$

$$\mathbf{b}_l(t) = \mathbf{p}_l(t) \odot \mathbf{e}_l(t) \tag{10}$$

where $\tau_u$ is the membrane leak time constant of the apical dendrite. Finally, the feedforward weights change over time following:

$$\frac{d\mathbf{W}_l(t)}{dt} = \frac{1}{\tau_W}\Big(\Big[\mathbf{p}_l(t) - \mathbf{p}_l^b\Big] \odot \mathbf{e}_l(t)\Big)\mathbf{e}_{l-1}^T(t) \tag{11}$$

where $\tau_W$ is a time constant that determines the learning rate.

### A.2   Spiking BurstCCN

Previous backprop-like learning models rely on computations across distinct phases to obtain the necessary error signals to learn [10, 27]. For example, Burstprop, proposed by Payeur et al. [10], computes the difference between the level of bursting between two phases: an initial phase with no teaching signal and a second with a teaching signal present. The burst-dependent plasticity rule used in the spiking implementation of Burstprop requires neuron ensembles to compute a moving average of their burst probability over time. This quantity needs to reach stability in order to generate useful error signals. Therefore, there must be a period with no plasticity before a teaching signal is presented. This requires global plasticity switches between the two phases which is problematic because a high level of coordinated plasticity is needed across the whole network. To address these issues, we build upon Burstprop's plasticity rule and propose a spiking implementation of our rate-based burst-dependent plasticity rule (Eq. 1):

$$\frac{dw_{ij}}{dt} = \eta\Big[B_i(t) - P_i^b E_i(t)\Big]\tilde{E}_j(t) \tag{12}$$

where $P^b$ is the baseline burst probability, $\tilde{E}$ is an eligibility trace of presynaptic activity (with time constant $\tau_{pre}$), $\eta$ is a learning rate and $B_i$ and $E_i$ are burst and event trains, respectively. In contrast to Burstprop, we use this constant baseline burst probability ($P^b$) instead of a moving average of the burst probability to control the relative magnitude of positive and negative weight changes. This makes it unnecessary to have a separate phase to compute its value since it is no longer time-dependent. However, to make single phase learning possible, we have also introduced a set of STD feedback weights that cancel baseline bursting feedback activity into apical compartments such that only deviations from the baseline are communicated. With just STF feedback weights, as is the case for Burstprop, cells will not consistently burst at baseline in the absence of a teaching signal.

This is due to the bursting signals they communicate having a dependence on the feedforward activity (i.e. events) and necessarily varying across input stimuli.

We include this spike-based burst-dependent synaptic plasticity rule in a spiking implementation of the BurstCCN. Unlike the two rate-based implementations, the spiking BurstCCN more accurately models the internal neuron spiking dynamics instead of abstracting these details away and only considering the ensemble-level behaviour. Pyramidal neurons are modelled with two compartments corresponding to the soma and apical dendrites. The remaining details of this model along with the hyperparameters used are described by Payeur et al. [10] and the simulation of all neurons was carried out using the Auryn simulator [28].

## B   BurstCNN approximates the backpropagation algorithm

In Section 3.3.1, we discussed the relationship between BurstCNN and the error backpropagation algorithm. Here, we demonstrate their relationship formally and specify the constraints required for an equivalence.

We first remind the reader of the process of error backpropagation. Using the notation of the main text and letting $\mathbf{g}_l = \frac{\partial E^{\text{task}}}{\partial \mathbf{v}_l}$ denote the error gradient with respect to the somatic potentials at layer $l$, the error backpropagation algorithm generates $\mathbf{g}_l$ recursively with

$$\mathbf{g}_L = f'\left(\mathbf{v}_L\right) \odot \frac{\partial E^{\text{task}}}{\partial \mathbf{e}_L} \tag{13}$$

$$\mathbf{g}_l = f'\left(\mathbf{v}_l\right) \odot \mathbf{W}_{l+1}^T \mathbf{g}_{l+1} \tag{14}$$

where $f'$ denotes the first order derivative of $f$. The error gradients with respect to the feedforward weights onto layer $l$ can then simply be obtained with $\frac{\partial E^{\text{task}}}{\partial \mathbf{W}_l} = \frac{\partial E^{\text{task}}}{\partial \mathbf{v}_l}\left(\frac{\partial \mathbf{v}_l}{\partial \mathbf{W}_l}\right)^T = \mathbf{g}_l \mathbf{e}_{l-1}^T$.

To demonstrate an equivalence of the weight updates obtained by BurstCNN (Eq. 1) with those of error backpropagation, it is necessary to show that the difference between the burst rate and its baseline $\delta \mathbf{b}_l := \left(\mathbf{p}_l - \mathbf{p}_l^b\right) \odot \mathbf{e}_l$ mirrors the quantity $-\mathbf{g}_l$ (negated so that increases in bursting lead to LTP). Note that by construction this is true (up to the proportionality constant $p_l^b$) for the output layer $L$ since

$$\begin{aligned}
\delta \mathbf{b}_L &= \mathbf{p}_L^b \odot \left(\mathbf{e}_{target} - \mathbf{e}_L\right) \odot h(\mathbf{e}_L) \odot \mathbf{e}_L \\
&= p_L^b f'\left(\mathbf{v}_l\right) \odot \left(\mathbf{e}_{target} - \mathbf{e}_L\right) \\
&= p_L^b f'\left(\mathbf{v}_l\right) \odot \left(-\frac{\partial E^{\text{task}}}{\partial \mathbf{e}_l}\right) \\
&= -p_L^b \mathbf{g}_L.
\end{aligned}$$

To demonstrate that $\delta \mathbf{b}_l \approx -p_L^b \mathbf{g}_l$ for layers $l < L$ it is then left to show that $\delta \mathbf{b}_l$ satisfies the recursive relationship in Equation 14. We proceed under the assumption of the $\mathbf{Q}$-$\mathbf{Y}$ symmetric state, $\mathbf{Q}_l = p_l^b \mathbf{Y}_l$, under the particular case that $p_l^b = \frac{1}{2}$ for each layer $l$. We also choose $\bar{\sigma}(x) = \sigma(4x)$ (i.e. $\alpha = 4$ and $\beta = 0$; see Section 2.2). By applying the definition of $\delta \mathbf{b}_l$ and expanding the sigmoid term in the burst probability $\mathbf{p}_l$ by its Taylor series, $\sigma(x) = \frac{1}{2} + \frac{x}{4} + \frac{x^3}{48} + \ldots$, we obtain

$$\delta\mathbf{b}_l = (\mathbf{p}_l - p_l^b) \odot \mathbf{e}_l$$

$$= (\mathbf{p}_l - \frac{1}{2}) \odot \mathbf{e}_l \qquad\qquad \text{(since } p_l^b = \frac{1}{2})$$

$$= \mathbf{p}_l \odot \mathbf{e}_l - \frac{1}{2}\mathbf{e}_l$$

$$= \left(\bar{\sigma}\left(h(\mathbf{e}_l) \odot \mathbf{u}_l\right)\right) \odot \mathbf{e}_l - \frac{1}{2}\mathbf{e}_l$$

$$= \left(\sigma\left(4h(\mathbf{e}_l) \odot \mathbf{u}_l\right)\right) \odot \mathbf{e}_l - \frac{1}{2}\mathbf{e}_l \qquad\qquad \text{(since } \bar{\sigma}(\mathbf{u}_l) = \sigma(4\mathbf{u}_l))$$

$$= \left(\frac{1}{2} + \frac{4h(\mathbf{e}_l) \odot \mathbf{u}_l}{4} + \mathcal{O}(\mathbf{u}_l^3)\right) \odot \mathbf{e}_l - \frac{1}{2}\mathbf{e}_l \qquad\qquad \text{(Taylor series expansion)}$$

$$= \frac{1}{2}\mathbf{e}_l + h(\mathbf{e}_l) \odot \mathbf{u}_l \odot \mathbf{e}_l + \mathcal{O}(\mathbf{u}_l^3) - \frac{1}{2}\mathbf{e}_l$$

$$= h(\mathbf{e}_l) \odot (\mathbf{Q}_l\mathbf{e}_{l+1} - \mathbf{Y}_l\mathbf{b}_{l+1}) \odot \mathbf{e}_l + \mathcal{O}(\mathbf{u}_l^3) \qquad\qquad \text{(by definition of } \mathbf{u}_l)$$

$$= h(\mathbf{e}_l) \odot \left(\frac{\mathbf{Y}_l}{2}\mathbf{e}_{l+1} - \mathbf{Y}_l\mathbf{b}_{l+1}\right) \odot \mathbf{e}_l + \mathcal{O}(\mathbf{u}_l^3) \qquad\qquad \text{(since } \mathbf{Q}_l = p_l^b\mathbf{Y}_l)$$

$$= h(\mathbf{e}_l) \odot \mathbf{e}_l \odot (-\mathbf{Y}_l)\left(\mathbf{b}_{l+1} - \frac{1}{2}\mathbf{e}_{l+1}\right) + \mathcal{O}(\mathbf{u}_l^3)$$

$$= h(\mathbf{e}_l) \odot \mathbf{e}_l \odot (-\mathbf{Y}_l)\left(\left(\mathbf{p}_{l+1} - \frac{1}{2}\right) \odot \mathbf{e}_{l+1}\right) + \mathcal{O}(\mathbf{u}_l^3) \quad \text{(since } \mathbf{b}_{l+1} = \mathbf{p}_{l+1} \odot \mathbf{e}_{l+1})$$

$$= f'(\mathbf{v}_l) \odot (-\mathbf{Y}_l)\left(\left(\mathbf{p}_{l+1} - \frac{1}{2}\right) \odot \mathbf{e}_{l+1}\right) + \mathcal{O}(\mathbf{u}_l^3) \qquad \text{(since } f'(\mathbf{v}_l) = h(\mathbf{e}_l) \odot \mathbf{e}_l)$$

$$= f'(\mathbf{v}_l) \odot (-\mathbf{Y}_l)\left(\left(\mathbf{p}_{l+1} - p_{l+1}^b\right) \odot \mathbf{e}_{l+1}\right) + \mathcal{O}(\mathbf{u}_l^3)$$

$$= f'(\mathbf{v}_l) \odot (-\mathbf{Y}_l)\delta\mathbf{b}_{l+1} + \mathcal{O}(\mathbf{u}_l^3).$$

Here, we use the abuse of notation $\mathbf{u}_l^3 = (u_{l,1}^3, u_{l,2}^3, \dots)^T$ and apply the Taylor series expansion under the assumptions that the apical potential is bounded by 1, $||\mathbf{u}_l||_\infty \le 1$ (so that the cube term dominates), and that $h$ is bounded (so that the expanded term $h(\mathbf{e}_l) \odot \mathbf{u}_l^3$ is of order $\mathcal{O}(\mathbf{u}_l^3)$). Note that for the sigmoid forward activation $f = \sigma$ the latter is satisfied automatically since $h(\mathbf{e}_l) = 1 - \mathbf{e}_l$ with $\mathbf{e}_l$ itself bounded (between 0 and 1). The above derivation then leads to the result cited in the main text (Eq. 3). In particular, if the apical potential is small enough such that $\mathcal{O}(\mathbf{u}_l^3) \approx 0$ and the $\mathbf{W}$ and $\mathbf{Y}$ weights are aligned with $\mathbf{Y}_l = -\mathbf{W}_{l+1}^T$, we have an equivalence to the error backpropagation algorithm of Equation 14:

$$\delta\mathbf{b}_l \approx f'(\mathbf{v}_l) \odot \mathbf{W}_{l+1}^T \delta\mathbf{b}_{l+1} \qquad\qquad (15)$$

## B.1 Backpropagation of apical potentials

In this section, we show that if the Euclidean norm of the cube of the apical potentials, $||\mathbf{u}_l^3||_2$, is small at the output layer and the feedback weights are weak then by induction it remains small across all layers. This demonstrates that the approximation error in Equation 3 also remains small across layers as the $\mathbf{u}_l^3$ term is a dominant factor when $\mathbf{u}_l$ is small. We again assume that $p_l^b = \frac{1}{2}$ and that the weights are in the $\mathbf{Q}$-$\mathbf{Y}$ symmetric regime, $\mathbf{Q}_l = \frac{\mathbf{Y}_l}{2}$. The apical potential at layer $l$ can then be written as

$$\mathbf{u}_l = \mathbf{Q}_l \mathbf{e}_{l+1} - \mathbf{Y}_l \mathbf{b}_{l+1}$$
$$= \frac{\mathbf{Y}_l}{2}\mathbf{e}_{l+1} - \mathbf{Y}_l\left(\mathbf{p}_{l+1} \odot \mathbf{e}_{l+1}\right)$$
$$= \frac{\mathbf{Y}_l}{2}\mathbf{e}_{l+1} - \mathbf{Y}_l\left(\sigma(4h(\mathbf{e}_{l+1}) \odot \mathbf{u}_{l+1}) \odot \mathbf{e}_{l+1}\right)$$
$$= \frac{\mathbf{Y}_l}{2}\mathbf{e}_{l+1} - \mathbf{Y}_l\left(\left(h(\mathbf{e}_{l+1}) \odot \mathbf{u}_{l+1} + \frac{1}{2}\right) \odot \mathbf{e}_{l+1}\right)$$
$$= \frac{\mathbf{Y}_l}{2}\mathbf{e}_{l+1} - \mathbf{Y}_l h(\mathbf{e}_{l+1}) \odot \mathbf{u}_{l+1} \odot \mathbf{e}_{l+1} - \frac{\mathbf{Y}_l}{2}\mathbf{e}_{l+1}$$
$$= (-\mathbf{Y}_l)(h(\mathbf{e}_{l+1}) \odot \mathbf{e}_{l+1} \odot \mathbf{u}_{l+1})$$
$$= (-\mathbf{Y}_l)(f'(\mathbf{v}_{l+1}) \odot \mathbf{u}_{l+1}).$$

Intuitively, given that $f'$ is bounded by 1 (as is the case for ReLU, sigmoid and tanh), we only require that the feedback weights $\mathbf{Y}_l$ (or equivalently the forward weights $\mathbf{W}_l$ in the $\mathbf{W}$-$\mathbf{Y}$ symmetric state) are small enough such that the cube of the final term above has a smaller or equal magnitude to the cube of the apical potentials in the next layer, $\mathbf{u}_{l+1}^3$. In fact, to formally ensure that the size of this term shrinks layer by layer, i.e. $||\mathbf{u}_l^3||_2 \leq ||\mathbf{u}_{l+1}^3||_2$, we can enforce the condition that the 6-norm of the feedback weights $\mathbf{Y}_l$ is at most 1; that is, $||\mathbf{Y}_l||_6 = \sup_{\mathbf{x} \neq 0} \frac{||\mathbf{Y}_l\mathbf{x}||_6}{||\mathbf{x}||_6} \leq 1$. With this we have

$$||\mathbf{u}_l^3||_2 = ||\mathbf{u}_l^3||_2$$
$$= (||\mathbf{u}_l||_6)^3$$
$$= (||\mathbf{Y}_l(f'(\mathbf{v}_{l+1}) \odot \mathbf{u}_{l+1})||_6)^3$$
$$\leq (||f'(\mathbf{v}_{l+1}) \odot \mathbf{u}_{l+1}||_6)^3$$
$$= (||\mathbf{u}_{l+1}||_6)^3$$
$$= ||\mathbf{u}_{l+1}^3||_2.$$

Hence we have that the magnitude of the cube of apical potentials at layer $l$ is smaller than or equal to those in layer $l+1$, $||\mathbf{u}_l^3||_2 \leq ||\mathbf{u}_{l+1}^3||_2$. Finally, we note that the uppermost defined apical potentials (at layer $L-1$) can be derived as

$$\mathbf{u}_{L-1} = \mathbf{Q}_{L-1}\mathbf{e}_L - \mathbf{Y}_{L-1}\mathbf{b}_L$$
$$= \frac{\mathbf{Y}_{L-1}}{2}\mathbf{e}_L - \mathbf{Y}_{L-1}\left(\frac{1}{2}\mathbf{e}_L - \frac{1}{2}h(\mathbf{e}_L) \odot \mathbf{e}_L \odot \frac{\partial E^{\text{task}}}{\partial \mathbf{e}_L}\right)$$
$$= f'(\mathbf{v}_L) \odot (-\mathbf{Y}_{L-1})\left(-\frac{1}{2}\frac{\partial E^{\text{task}}}{\partial \mathbf{e}_L}\right).$$

Although it is not explicitly defined in the BurstCCN, we can think of the apical potentials at the final layer $L$ as (at least computationally equivalent to) $-\frac{1}{2}\frac{\partial E^{\text{task}}}{\partial \mathbf{e}_N}$. If these are appropriately small, then by induction, all apical potentials of the lower layers $\mathbf{u}_{1 \leq l \leq L-1}$ will be appropriately small such that we have $\delta\mathbf{b}_l \approx -\frac{1}{2}\mathbf{g}_l$ for each $l$ and an equivalence to the weight updates defined in backpropagation (Equations 4, 5).

**Case where $p_l^b \neq \frac{1}{2}$**

Note that for $p_l^b \neq \frac{1}{2}$, a similar derivation as in Equation 3 can be made by applying an offset $\beta$ to the sigmoid $\bar{\sigma}(x) = \sigma(4 \cdot x + 4\beta)$ where $\beta = p_l^b - \frac{1}{2}$, in order to linearise the function around $-\beta$. However, this introduces an extra error term in the approximation which scales with $|p_l^b - \frac{1}{2}|$ and makes very high or low baseline burst probabilities undesirable.

### B.2 Local plasticity rules ensure Q-Y symmetry in absence of teaching signal

We now show that the local update rule for the STD feedback weights, $\mathbf{Q}$, as defined in Equation 2, leads to the $\mathbf{Q}$-$\mathbf{Y}$ symmetric state as required in the sections above. In particular, we show that in the absence of an explicit teaching signal at the output layer and assuming independent somatic potentials in the population, $\mathbf{Q}_l$ will converge onto the solution $\mathbf{Q}_l = p_l^b \mathbf{Y}_l$.

First, we note that Equation 2 is simply implementing gradient descent with respect to the size of the apical potential $||\mathbf{u}_l||_2$.

$$\Delta\mathbf{Q}_l \propto -\frac{\partial ||\mathbf{u}_l||^2}{\partial \mathbf{Q}_l} = -\frac{\partial ||\mathbf{Q}_l \mathbf{e}_{l+1} - \mathbf{Y}_l \mathbf{b}_{l+1}||^2}{\partial \mathbf{Q}_l} = -\left(\mathbf{Q}_l \mathbf{e}_{l+1} - \mathbf{Y}_l \mathbf{b}_{l+1}\right) \mathbf{e}_{l+1}^T = -\mathbf{u}_l \mathbf{e}_{l+1}^T \quad (16)$$

Following this gradient (with appropriate learning rate $\eta_l^{(\mathbf{Q})}$) should therefore ensure reaching a local minimum on $||\mathbf{u}_l||_2$. However, since this is simply a (linear) least squares regression problem, the function $\phi : \mathbf{Q} \to ||\mathbf{u}_l||_2^2$ is convex and therefore any local minimum is the global minimum.

To show that $\mathbf{Q}_l$ will converge onto $p_l^b \mathbf{Y}_l$ itself (rather than another solution), we see that, in the absence of a teaching signal, this solution enforces $\mathbf{u}_l = \mathbf{0}$.

$$\mathbf{u}_l = \mathbf{Q}_l \mathbf{e}_{l+1} - \mathbf{Y}_l \mathbf{b}_{l+1} = p_l^b \mathbf{Y}_l \mathbf{e}_{l+1} - p_l^b \mathbf{Y}_l \mathbf{e}_{l+1} = \mathbf{0} \quad (17)$$

Thus, $\mathbf{Q}_l = p_l^b \mathbf{Y}_l$ achieves the global minimum $||\mathbf{u}_l||_2^2 = ||0||_2^2 = 0$. Moreover, in the case that the event rates $\mathbf{e}_{l+1}$ are linearly independent, we have $\mathbf{Q}_l = p_l^b \mathbf{Y}_l$ as the unique solution. Although linear independence of $\mathbf{e}_{l+1}$ is not assured a priori, this can be guaranteed, for example, by independently injecting noise current into the population (see Section C.3.2). The (global) minimum obtained by the update rules in Equation 2 must therefore correspond to this solution $\mathbf{Q}_l = p_l^b \mathbf{Y}_l$, ensuring $\mathbf{Q}$-$\mathbf{Y}$ symmetry as required.

## C  Experimental details

In the following section, we provide additional details on the experimental setups used to obtain the results in the main text.

### C.1  Spiking XOR task

In the spiking XOR task, we used networks consisting of 5 distinct neuron populations: 2 populations that encoded each input, 2 hidden layer populations and a single population that encoded the output. Each population consisted of 500 individual neurons. These neurons were sparsely connected in the feedforward and feedback directions with a connection probability of $0.05$. All neurons received balanced excitatory and inhibitory Poisson noise into both their somatic and dendritic compartments.

In the two-phase setup, learning rates for the weights and biases were set to $\eta_W = 0.004$ and $\eta_B = 0.0001$, respectively. We reduced learning rates to $\eta_W = 0.0004$ and $\eta_B = 0.00001$ in the one-phase setup due to the increased duration where plasticity was on.

Each simulation was carried out for $16000s$ (2000 examples $\times$ $8s$ per example) with timesteps of $dt = 0.1s$. Learning of the feedforward weights was carried out by each model's burst-dependent synaptic plasticity rule with the time constant of the pre-synaptic input eligibility trace $\tau_{pre} = 0.1s$. The moving average time constant for Burstprop, $\tau_{ma}$, was set to $2s$. Baseline burst probabilities in the BurstCCN were set to $P_{hidden}^b = 0.18$ and $P_{out}^b = 0.401$ in the hidden and output layers, respectively. The remaining task details can be found in [10].

### C.2  Continuous-time input-output task

This task was carried out using a 2-layer continuous-time BurstCCN with 3 inputs, 50 hidden units and a single output unit. The simulation was ran for $10^6$ timesteps with $dt = 0.1s$ to give a total simulation time of $10^5 s$. During the first $100s$ of the simulation, plasticity and teaching signals

were switched off to provide a means of comparing the initialised network to the trained network (Fig. 3C-D). The time constants used in the simulation for the somatic potentials, dendritic potentials and the synaptic weights were set to $\tau_v = 0.1s$, $\tau_u = 0.1s$ and $\tau_W = 100.0s$, respectively.

## C.3 Rate-based model experiments

### C.3.1 Hyperparameter search

Bayesian hyperparameter optimisation was performed (using Weights & Biases [29]) for the shared hyperparameters of each rate-based model shown in Tables S1-S2 and S4-S5. This was also performed for the model-specific parameters that are stated in the following relevant sections.

### C.3.2 Feedback plasticity on Q weights

As described in Section 3.3.2, this task involved fixing the $\mathbf{W}$ and $\mathbf{Y}$ weights and allowing the $\mathbf{Q}$ weights to learn following the plasticity rule given in Equation 2. Throughout training, independent Gaussian noise was added to the event rates propagating forwards to give somatic potentials, $\mathbf{v}_l = \mathbf{W}_l(\mathbf{e}_{l-1} + \xi)$ where $\xi \sim \mathcal{N}(\mathbf{0}, \sigma^2\mathbf{I})$. This noise was introduced to decorrelate neural activities in each layer to facilitate the alignment of $\mathbf{Q}$ to $\mathbf{Y}$ (see Section B.2). For all configurations, we set $\sigma = 0.1$ during training but removed the noise when evaluating the apical potential magnitudes (Fig. 4B,F) and alignment angles (Fig.4C-D,G-H). Similarly, in the without-teacher setups, teaching signals were added only for the evaluation of the alignment angles to backprop or feedback alignment. Table S1 shows the best hyperparameters for each setup optimised for $\mathbf{Q}$-$\mathbf{Y}$ alignment.

Table S1: Q-weight learning hyperparameters

| Feedback mode | Teacher | Q learning rate ($\eta^{(\mathbf{Q})}$) | Y init. scale ($\sigma_{\mathbf{Y}}$) | Q init. scale ($\sigma_{\mathbf{Q}}$) |
|---|---|---|---|---|
| Random (FA) | No | 0.00520 | 0.5 | 0.0148 |
|  | Yes | 0.00380 | 0.5 | 0.0488 |
| Symmetric | No | 0.00337 | N/A | 0.0075 |
|  | Yes | 0.00677 | N/A | 0.0735 |

### C.3.3 MNIST

For MNIST, we used standard ANN, BurstCCN, Burstprop and EDN models. The same network structure was used for all model types: 784 inputs ($28 \times 28$ grayscale images), $n$ number of hidden layers with 500 units and 10 output units. In all cases, we used a sigmoid for the feedforward activation functions and a batch size of 32. Table S2 shows the best hyperparameters found for the 5-layer networks of each model which were shared across all network sizes.

**BurstCCN** The additional $\mathbf{Q}$ weights were trained with a learning rate of $3.5 \times 10^{-5}$ and the baseline burst probability $\mathbf{p}_l^b = 0.5$ for all layers.

**Burstprop** The additional recurrent weights were initialised from a Gaussian distribution (with $\mu = 0$ and $\sigma = 1.3 \times 10^{-4}$) and trained with a learning rate of $2.6 \times 10^{-4}$. The baseline burst probability at the output layer was set to 0.2.

**EDN** Learning rates for the interneuron-to-pyramidal ($\eta_{l,l}^{\mathrm{PI}}$) and pyramidal-to-interneuron ($\eta_{l,l}^{\mathrm{IP}}$) weights were set proportionally to the feedforward learning rates ($\eta_{l+1,l}^{\mathrm{PP}}$) as $\eta_{l,l}^{\mathrm{PI}} = \eta_{l,l}^{\mathrm{IP}} = \alpha\eta_{l+1,l}^{\mathrm{PP}}$ with the proportionality constant $\alpha = 0.643$.

### C.3.4 CIFAR-10

For CIFAR-10, we used convolutional ANN, BurstCCN and Burstprop models with three convolutional layers followed by two fully-connected layers (Table S3). In all cases, we used a sigmoid for the feedforward activation functions and a batch size of 32. Table S4 and Table S5 show the best hyperparameters shared across the different models in the random fixed $\mathbf{Y}$ and $\mathbf{W}$-$\mathbf{Y}$ symmetric feedback weight regimes, respectively.

Table S2: MNIST hyperparameters

| Model Type | Learning rate ($\eta$) | Y init. scale ($\sigma_{\mathbf{Y}}$) | Momentum | Weight decay |
|---|---|---|---|---|
| ANN | 0.201 | 1.49 | 0.474 | $1.09 \times 10^{-9}$ |
| BurstCCN | 0.0246 | 0.638 | 0.836 | $4.01 \times 10^{-10}$ |
| BurstCCN ($\mathbf{Q}$-$\mathbf{Y}$ sym) | 0.0246 | 0.638 | 0.836 | $4.01 \times 10^{-10}$ |
| Burstprop | 0.481 | 1.71 | 0.347 | $7.59 \times 10^{-6}$ |
| EDN | 0.00578 | 2.60 | 0.198 | $2.01 \times 10^{-7}$ |

**BurstCCN** The additional $\mathbf{Q}$ weights were trained with a learning rate of $2.21 \times 10^{-3}$ and the baseline burst probability $\mathbf{p}_l^b = 0.5$ for all layers.

**Burstprop** Under the random fixed $\mathbf{Y}$ feedback weight regime, the additional recurrent weights were initialised from a Gaussian distribution (with $\mu = 0$ and $\sigma = 2.0 \times 10^{-4}$) and trained with a learning rate of $2.0 \times 10^{-4}$. Under the $\mathbf{W}$-$\mathbf{Y}$ symmetric weight regime, the additional recurrent weights were initialised from a Gaussian distribution (with $\mu = 0$ and $\sigma = 2.5 \times 10^{-2}$) and trained with a learning rate of $2.2 \times 10^{-5}$. For both weight regimes, the baseline burst probability at the output layer was set to 0.2.

Table S3: CIFAR-10 architecture

| Layer number | Layer type | Size |
|---|---|---|
| 0 | Input | $32 \times 32 \times 3$ |
| 1 | Convolutional | $5 \times 5$, 64, stride=2 |
| 2 | Convolutional | $5 \times 5$, 128, stride=2 |
| 3 | Convolutional | $3 \times 3$, 256, stride=1 |
| 4 | Fully-connected | 1480 neurons |
| 5 | Fully-connected | 10 neurons |

Table S4: CIFAR-10 hyperparameters (random fixed $\mathbf{Y}$ feedback weight regime)

| Model Type | Learning rate ($\eta$) | Y init. scale ($\sigma_{\mathbf{Y}}$) | Momentum | Weight decay |
|---|---|---|---|---|
| ANN | 0.00335 | 0.363 | 0.502 | $5.96 \times 10^{-8}$ |
| BurstCCN | 0.0343 | 0.424 | 0.0142 | $1.01 \times 10^{-10}$ |
| BurstCCN ($\mathbf{Q}$-$\mathbf{Y}$ sym) | 0.0132 | 0.290 | 0.578 | $2.65 \times 10^{-9}$ |
| Burstprop | 0.102 | 0.981 | 0.698 | $3.91 \times 10^{-9}$ |

# D  Additional experiments

Here we describe a set of additional experiments that we have conducted. The first demonstrates that the distal cancellation can also be performed by having plasticity on the $\mathbf{Y}$ weights. The second shows the results for MNIST in the $\mathbf{W}$-$\mathbf{Y}$ symmetric regime which more closely resembles backprop. Finally, we use an idealised version of the EDN to highlight the cause of the EDN's poor performance.

## D.1  Feedback plasticity on Y-weights

As stated in the main text, the feedback plasticity rule used on the $\mathbf{Q}$ weights could likewise be applied to the $\mathbf{Y}$ weights to produce a similar effect. This would be in line with long-term synaptic plasticity observations of dendrite-targeting interneurons [30].

$$\Delta \mathbf{Y}_l = \eta_l^{(\mathbf{Y})} \, \mathbf{u}_l \, \mathbf{e}_{l+1}^T \tag{18}$$

In an identical experimental setup to the one used for investigating the feedback plasticity rule on the $\mathbf{Q}$ weights (Section 3.3.2), we examined the impact on the alignment to backprop when the feedback plasticity rule was instead applied onto the $\mathbf{Y}$ weights. In all cases, as the alignment

| Model Type | Learning rate ($\eta$) | Momentum | Weight decay |
|---|---|---|---|
| ANN | 0.0814 | 0.447 | $1.60 \times 10^{-6}$ |
| BurstCCN ($\mathbf{Q}$-$\mathbf{Y}$ sym) | 0.155 | 0.496 | $1.50 \times 10^{-8}$ |
| Burstprop | 0.146 | 0.869 | $2.53 \times 10^{-8}$ |

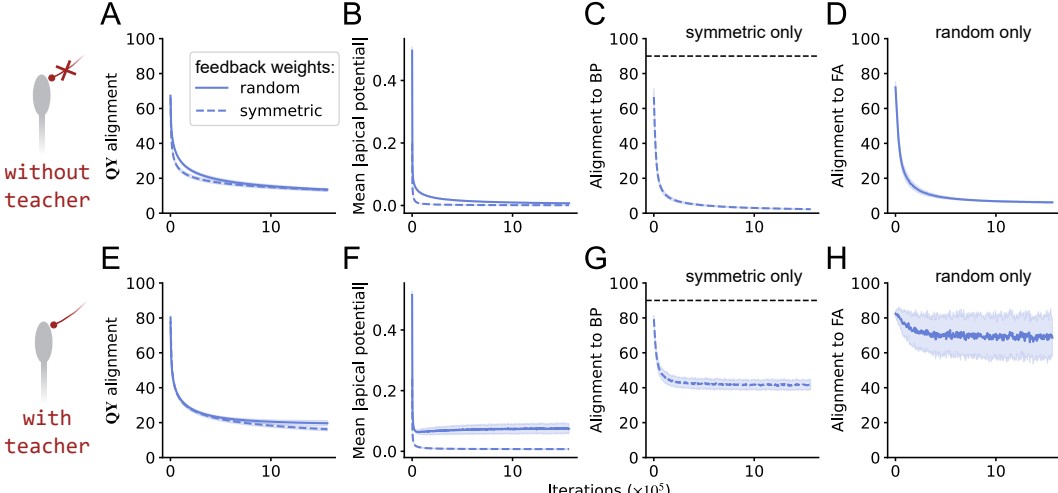

Figure S1: **Feedback plasticity rule on $\mathbf{Y}$ weights facilitates updates to align with backprop and feedback alignment.** The network is a randomly initialised 5-layer discrete-time BurstCCN with random (solid line) or symmetric (dashed line), fixed $\mathbf{W}$ and $\mathbf{Y}$ weights. The $\mathbf{Y}$ weights are updated in the presence of (**A-D**) no teaching signal or (**E-H**) a teaching signal. (A,E) Alignment between $\mathbf{Q}$ and $\mathbf{Y}$ connections, (B,F) the mean absolute value of the apical potentials, (C,G) the alignment to backprop (BP) and (D,H) feedback alignment (FA) as $\mathbf{Y}$ weights learn to silence apical dendrite potential. Updates below $90°$ marked by the black dashed line are considered useful as they still follow the direction to backprop on average. Model results represent mean $\pm$ standard error (n = 5).

between the $\mathbf{Q}$ and $\mathbf{Y}$ connections improved (Fig. S1A,E), the apical potential decreased (Fig. S1B,F) and this resulted in updates that more closely aligned to backprop (Fig. S1C,G) and feedback alignment (Fig. S1D,H). In the absence of a teaching signal, this alignment angle to both backprop and feedback alignment eventually became very small (Fig. S1C-D). Despite producing less aligned feedforward updates in the presence of a teaching signal, the updates computed were still informative since they were consistently well below $90°$ of the direction of steepest descent (Fig. S1G). Overall, this suggests that the feedback plasticity rule on the $\mathbf{Q}$ weights is equally effective when applied to the $\mathbf{Y}$ weights.

## D.2  Symmetric MNIST

Here, we trained the same models used in Figure 5 under the $\mathbf{W}$-$\mathbf{Y}$ symmetric weight regime. This regime is implausible due to sharing of weights but we investigated this to isolate the ability of each network to backpropagate errors in the ideal setting that most resembles backprop.

Using 5-layer networks, BurstCCN ($\mathbf{Q}$-$\mathbf{Y}$ sym) obtained a test error of 1.86$\pm$0.03% similar to that of an ANN (1.97$\pm$0.04%) and outperforming both Burstprop (2.79$\pm$0.03%) and EDN (12.28$\pm$0.26%) (Fig. S2A). However, the BurstCCN with $\mathbf{Q}$ weight learning achieved a performance of 5.27$\pm$0.07%. This was lower than the random fixed $\mathbf{Y}$ weight regime (Fig. 5) due to the increased difficulty of aligning with the rapidly moving $\mathbf{Y}$ weights as the network learns. As the network depth was increased, both BurstCCN and Burstprop retained high performances but the EDN showed a substantial decay in performance with deeper networks (Fig. S2B). We then compared the alignment between the models and backprop. For the 5-layer networks, BurstCCN consistently produced updates that aligned with backprop at around $50°$ (Fig. S2C). As expected, the BurstCCN with $\mathbf{Q}$-$\mathbf{Y}$

symmetry was significantly better at propagating error signals. With increased network depths, we demonstrate that it was more difficult to produce updates that were closely aligned to backprop. However, we show that the BurstCCN was still capable of backpropagating useful error signals in relatively deep networks (Fig. S2D).

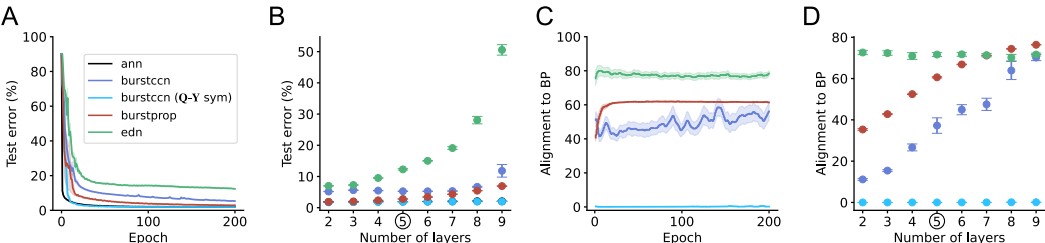

Figure S2: **BurstCCN with W-Y symmetric weights.** (**A**) Learning curve and (**C**) alignment to backprop of 5-layer BurstCCN (blue), BurstCCN ($\eta^{(\mathbf{Q})} = 0$) (light blue), Burstprop (red) and EDN (green). (**B**) Different number of hidden layers across all models. (**D**) Alignment to backprop (BP) across number of hidden layers. The black circle indicates that the hyperparameters for each model were optimised for 5-layer networks. Model results represent mean $\pm$ standard error (n = 5).

## D.3    EDN with interneuron weight symmetry

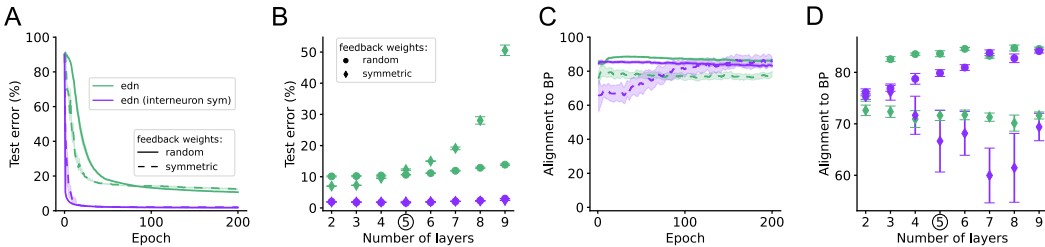

Figure S3: **EDN with interneuron symmetric weights.** (**A**) Learning curve of 5-layer EDN (green) and EDN in the interneuron symmetric weight regime (purple) with random (solid) or symmetric (dashed) feedback weights. (**B**) Test error with different numbers of hidden layers for all models. (**C**) Alignment to backprop (BP) over time for all 5-layer models. (**D**) Alignment to backprop with different numbers of hidden layers for all models. Circles and diamonds indicate the random and symmetric feedback weight regimes, respectively. The black circle indicates that the hyperparameters for each model were optimised for 5-layer networks. Model results represent mean $\pm$ standard error (n = 5).

Each layer in the EDN includes a set of apical-targeting interneurons which predict next layer activity and facilitate the generation of error signals for learning. These interneurons have lateral connections to and from pyramidal cells in the same layer. In the ideal case, the weights of the connections to and from these interneurons should exactly mirror the feedforward and feedback connections between the layers of pyramidal cells, respectively. We refer to this as the interneuron symmetric weight regime. This exact symmetry is not biologically plausible so the standard EDN model uses local plasticity rules for these connections to learn approximate symmetries.

Here, we evaluated the performance of a standard EDN model and an EDN model in the interneuron symmetric weight regime on MNIST to examine the impact of the imperfect weight symmetries present in the standard model. Using 5-layer networks, we show that the EDN in the interneuron symmetric weight regime (random: 1.644±0.018%, symmetric: 1.962±0.019%) significantly outperformed the EDN (random: 12.28±0.26%, symmetric: 10.65±0.09%) (Fig. S3A). This was irrespective of the feedback weight regime demonstrating the importance of the interneuron weight symmetry. As the network depth increased, the EDN, particularly in the symmetric weight regime, suffered a substantial reduction in performance. However, this large effect of depth on performance was less evident with the EDN under the interneuron symmetric weight regime (Fig. S3B). As

expected, the alignment to backpropagation was closer with symmetric feedback weights compared to random fixed feedback weights (Fig. S3C-D).

## E   Compute resources

The experiments involving the Auryn simulator were conducted with an AMD Ryzen 7 3700X CPU. For the different methods, we ran a total of 5 seeds which each took approximately 3 hours. The remaining experiments were conducted using NVIDIA GeForce RTX 2080 Ti GPUs. The runtime for each experiment per seed was approximately:

- **Continuous input-output task** - 50 minutes ($10^6$ timesteps)
- **Q weight plasticity** - 6 hours (1000 epochs)
- **MNIST** - 50 minutes (200 epochs)
- **CIFAR-10** - 3 hours (400 epochs)

## F   Code availability

The source code for this project can be found at `https://github.com/neuralml/BurstCCN`.