# OpenReview forum: "Single-phase deep learning in cortico-cortical networks"
_NeurIPS.cc/2022/Conference — NeurIPS 2022 Accept_

### Official Review · Reviewer_BqpL · 2022-07-11

**Rating:** 7
**Confidence:** 3
**Soundness:** 3 good
**Presentation:** 3 good
**Contribution:** 3 good

**Summary:**

This paper introduces a biological learning algorithm in multi-layer neural networks that uses information in neurons' spike trains (specifically, the distinction between overall firing rate and frequency of burst firing) to multiplex forward-propagated signals and backward-propagated error signals.  This approach relies heavily on short-term plasticity in feedforward and feedback signals to transmit firing rate and bursting information independently, and is shown to approximately implement backpropagation under certain conditions.  Unlike prior work, the model is capable of learning in a single phase, rather than separate inference and supervision phases, and can also handle learning from time-varying input.



**Questions:**

My main suggestion for the authors would be to more clearly delineate the contributions of their work relative to BurstProp (this is done somewhat in the Appendix, but I think it warrants more treatment in the main text).

Also, as mentioned above, it would be helpful to clarify the relationship between the spiking and rate-based model.  I would also suggest further empirical evaluation of the model on another (larger) dataset.

I don't hold it against the paper for not including this, but it seems like it could be fairly straightforward to implement a version of the model with mechanisms to enforce weight symmetry (as mentioned in the conclusion section) that could improve performance.

**Limitations:**

The authors are clear and upfront about the limitations of the work.

**Strengths And Weaknesses:**

This paper addresses an important question -- how biological neural networks can usefully multiplex inference and learning signals without unrealistic switching between network phases.  As far as I know this work is the first to resolve this question in a reasonably biologically realistic way.  The presentation and derivations are clear, the mechanisms seem biologically plausible and have some grounding in evidence from neuroscience, and the experimental results are compelling.

One main potential weakness of this paper is that it is based heavily on a prior model, BurstProp.  Many of the key ideas of the model (multiplexing firing rate + burst probability information, short-term faciliation and depression to transmit signals independently) were present already in the BurstProp model.  The contributions of this paper enable single-phase learning, which I do believe is an important advance, but the paper would benefit from clarifying and emphasizing which features of the model are novel (and needed to enable single-phase learning) and which are derived from prior work.

Another concern I have is that the experiments are limited in scope and scale.  A variety of biologically plausible learning algorithms have been proposed in the literature that perform well on datasets like MNIST and CIFAR-10 but fail to scale to e.g. ImageNet, or to other non-image-classification tasks.

Finally, I find it a bit unclear what the precise relationship is between the spiking and rate-based implementations.  Are they exactly equivalent in some limit?  I am aware that training spiking networks is generally more difficult than training rate networks, so I don't have a problem with the use of the rate model for the more complicated tasks, but it would help to clarify the link between the two models.

---

> ### Author Response · Authors · 2022-08-02
> **Response to Reviewer BqpL**
>
> > **"One main potential weakness of this paper is that it is based heavily on a prior model [...] the paper would benefit from clarifying and emphasizing which features of the model are novel (and needed to enable single-phase learning) and which are derived from prior work.”**
>
> > **“My main suggestion for the authors would be to more clearly delineate the contributions of their work relative to BurstProp (this is done somewhat in the Appendix, but I think it warrants more treatment in the main text).**
>
> Thank you for the suggestion. Our model indeed builds on Burstprop, but also on EDNs (Sacramento et al. 2018 NeurIPS), bringing components of both together. For the camera-ready version, we will emphasise the novel features of our model throughout the main text and include an additional paragraph that reads as follows: “Within each layer, Burstprop includes a set of recurrent connections onto apical compartments which aim to maintain the dendritic potential in the linear regime of the feedback non-linearity. Updating the weights of these connections requires separate learning phases and it is unclear how the plasticity rule can be justified. In contrast, the BurstCCN does not require these connections. Instead, we have introduced a novel set of STD feedback connections onto the apical dendrites which provide a mechanism for single-phase learning and can perform a similar role of linearising the feedback. Unlike Burstprop, burst-dependent plasticity in our model relies on a constant baseline burst probability (see SM Section A.2 for more information).”
>
> >**“Another concern I have is that the experiments are limited in scope and scale. A variety of biologically plausible learning algorithms have been proposed in the literature that perform well on datasets like MNIST and CIFAR-10 but fail to scale to e.g. ImageNet, or to other non-image-classification tasks.”**
>
> We are interested in testing our model further on larger datasets like ImageNet but due to the time constraints for this submission we were not able to explore this. It is true that many biologically plausible models are capable of solving MNIST. However, we would argue that only a small number of models with a similar level of plausibility as ours are capable of solving CIFAR-10 with performances comparable to a standard ANN. Burstprop can achieve reasonable performance on ImageNet (56.1% top-5 error rate – see Fig. 6C in Payeur et al. 2021 Nature Neuroscience). Our results show that BurstCCN can outperform Burstprop (see Fig. 6), with a near-optimal alignment to backprop under ideal conditions (Q-Y and W-Y symmetry). Therefore, we believe that our model, under the right conditions, would be capable of achieving relatively good performance on ImageNet.
>
> >**“Finally, I find it a bit unclear what the precise relationship is between the spiking and rate-based implementations. Are they exactly equivalent in some limit? I am aware that training spiking networks is generally more difficult than training rate networks, so I don't have a problem with the use of the rate model for the more complicated tasks, but it would help to clarify the link between the two models.”**
>
> From spiking to continuous-time rate:
> Each unit of the continuous-time rate model represents a mean-field approximation of the spiking model’s neuron ensembles. The behaviour of these units is obtained by simplifying the neuron dynamics of the spiking model by ignoring the post-spike effects (e.g. spike-triggered adaptation) and replacing the spiking behaviour with a sigmoidal activation function to approximate the spike rate.
>
> From continuous-time rate to discrete-time rate:
> The updates made in the discrete-time rate model are equivalent to those made by the continuous-time rate model if each example input-output pair is presented until an equilibrium in dynamics is reached and a weight update for a single timestep is performed.
>
> >**“I don't hold it against the paper for not including this, but it seems like it could be fairly straightforward to implement a version of the model with mechanisms to enforce weight symmetry (as mentioned in the conclusion section) that could improve performance.”**
>
> It is straightforward to implement a learning rule for the feedback weights to learn to be symmetric with the feedforward weights. However, it is not clear how such a learning rule can be implemented in a biologically plausible way because simple solutions to learning this symmetry will require non-local information. This is something we are currently exploring and plan to include in future work. Indeed, Burstprop used the Kolen-Pollack algorithm (Akrout et al. 2020 arXiv) to learn the feedback weights, with promising results, which we would also expect to hold in our model. The reason we chose not to implement this algorithm is because it is not clear how such a rule can be implemented in biology.

---

> > ### Comment · Reviewer_BqpL · 2022-08-07
> > **Thanks for the response**
> >
> > Thanks for the thorough response to my comments.  In particular I find the proposed paragraph comparing the model to BurstProp to be a very useful addition.
> >
> > One additional comment that comes to mind based on your response is that it may be worth emphasizing in the text that the degree of biological plausibility of this model is, in fact, much greater than many other papers that propose "biologically plausible" deep learning algorithms.  A reader that does not appreciate this, but is familiar with say Akrout et al. 2020, might wonder what the contribution of this paper is.
> >
> > I maintain my recommendation of acceptance.

---

> > > ### Author Response · Authors · 2022-08-09
> > > **Response to Reviewer BqpL**
> > >
> > > Thank you for your comments. For the camera-ready version, we will make sure to add more emphasis in the text about the increased biological plausibility compared to other models.

---

### Official Review · Reviewer_niLx · 2022-07-11

**Rating:** 7
**Confidence:** 4
**Soundness:** 3 good
**Presentation:** 3 good
**Contribution:** 3 good

**Summary:**

The authors proposed a bursting cortico-cortical network which can perform backprop-like credit assignment in deep cortical networks. By combining bursting, short-term plasticity and dendrite-targeting inhibition, they showed that the model can effectively learn with a single learning phase, as well as in a continuous-time way.  They carried out both empirical and theoretical analyses to show that the weight learning approximates backprop-derived gradients. They also demonstrated the mode performance on learning complex image classification tasks on two benchmarks.

**Questions:**

Q1: why the feedforward weights are determiend in a short-term depression (STD) way while the feedback weights are determined in a joint way of STF and STD? Do these assumption have experimental support? STP are quite different across different brain regions and between different types of cell types (PC, PV SST, VIP, etc.). It seems there was a big assumption here.

Q2: In the dynamic input-output task, is the sinusoidal input fed into the network in a stream way?

Q3: When extending the model to a deep network (> 10 layers), how is the approximation of the weight update in the shallow layers with the proposed method? Does the approximation gradually become worse as the error signal propagates to the shallow layers?

Q4: Does the brain learn in a back-prop way, is there any direct evidence? If not, since deep neural networks with back-prop have already shown great performance in many ML tasks, why is the 'biological plausible deep learning' a problem?

**Limitations:**

Yes, the authors addressed the limitations of their work

**Strengths And Weaknesses:**

The paper is written clearly with a good logical flow. The question the authors tackled is instereting and may provide a solution for single-phase efficient deep learning in the brain (if the brain does learn in a back-prop way, see Q4 below). Both empirical and theoretical analyses are carried out to support the model. The model is a bit complex in its design but it seems novel.

---

> ### Author Response · Authors · 2022-08-02
> **Response to Reviewer niLx**
>
> >**“Q1: why the feedforward weights are determined in a short-term depression (STD) way while the feedback weights are determined in a joint way of STF and STD? Do these assumptions have experimental support? STP are quite different across different brain regions and between different types of cell types (PC, PV SST, VIP, etc.). It seems there was a big assumption here.”**
>
> The specific STD and STF connection types in the feedforward and feedback pathways are needed for the error backpropagation mechanism of our model (see Section 2.2). In brief, in the feedback pathways we require the STD connections to communicate a “without teacher” signal to compare with a “with teacher” signal from the STF connections which produces an error signal. The feedforward pathway has only STD connections because we do not want error information to be communicated forwards. These choices are also in line with experimental evidence. For example, Kinnischtzke et al. (2014) show STD on the feedforward cortical projections. A range of recent experimental studies show STF on cortical feedback onto SST-interneurons, and STD on pyramidal-to-pyramidal cortical feedback. We discuss this in lines 275-281. The fact that STP in our model is in line with experimental findings is one of the advantages of our model compared with previous models.
> We acknowledge that the STP on different connection types can vary across different brain regions but our choice of STP is very common throughout the brain and we feel that this extra biological feature would be of interest to study in future work.
>
> >**“Q2: In the dynamic input-output task, is the sinusoidal input fed into the network in a stream way?”**
>
> The sinusoidal input is fed into the network in a continuous manner. So, yes it is presented as a stream with each value of the wave being presented at each individual simulation timestep at increments of 0.1s rather than the entire sinusoidal input in a single timestep. We have now clarified that the input x is a time-varying function on line 177.
>
> >**“Q3: When extending the model to a deep network (> 10 layers), how is the approximation of the weight update in the shallow layers with the proposed method? Does the approximation gradually become worse as the error signal propagates to the shallow layers?”**
>
> We have not trained models with more than 10 layers but, yes, when using deep networks the alignment to backprop gradually gets worse for layers further from the output. This is a property of virtually all existing biological approximations of backprop. Each layer that the error backpropagates through introduces more deviation from the true backprop error due to the dendritic non-linearity and the degree of misalignment in feedback weights (both W-Y and Q-Y alignment). However, with perfect alignment of both feedback weights, our results show that our model can obtain a near-perfect approximation to backprop (Fig. 6C) and therefore we would expect it to scale well to deeper networks under these conditions.
>
> >**“Q4: Does the brain learn in a back-prop way, is there any direct evidence? If not, since deep neural networks with back-prop have already shown great performance in many ML tasks, why is the 'biological plausible deep learning' a problem?”**
>
> As of yet, there is no direct evidence that the brain learns specifically in a backprop-like way. The brain is clearly capable of learning a variety of tasks, which means that the updates to synaptic weights are descending a loss surface for those tasks. For the network structure we use, error backpropagation is the most efficient known method for computing the direction of steepest descent (i.e. the gradient). 'Biologically plausible deep learning' is a problem because it is currently unknown exactly how the brain can learn complex tasks. Deep learning has demonstrated the power of backprop but it is unclear how such a mechanism can be realised in biological circuitry. To clarify, our work does not aim to provide immediate benefits to the machine learning community but instead aims to further the understanding of how the brain could be learning. Additionally, researchers are beginning to experimentally test whether the brain does learn in a backprop-like way and our model provides predictions to be tested. The fact that we have a model that can learn challenging tasks while being consistent with a large number of biological features suggests that this is a promising direction forward.

---

> > ### Comment · Reviewer_niLx · 2022-08-09
> > **Thank you for the response**
> >
> > My concerns/questions are addressed. Congrats.

---

### Official Review · Reviewer_QqnL · 2022-07-11

**Rating:** 8
**Confidence:** 4
**Soundness:** 4 excellent
**Presentation:** 3 good
**Contribution:** 4 excellent

**Summary:**

The paper proposes a novel framework for biologically-plausible hierarchical learning in deep feedforward neural networks. Notably, they present an algorithm for credit assignment, that can approximate backprop, by leveraging certain known properties of cortical networks. The proposed algorithm leverages the dendritic segregation of neurons to have a two-compartment neuron model coupled with burst-dependent synaptic plasticity to perform credit assignment in a single phase. The authors do a commendable job in presenting the key components of their model, both for updating the feedforward weights as well as updating the dendritic feedback projections from higher layer neurons. Furthermore, they demonstrate that under specific assumptions, the proposed credit assignment rule can approximate backprop-based weight update. Finally, the algorithm was able to learn the XOR task and a non-linear regression task as well as achieved competitive performance on image classification task on MNIST and CIFAR-10 datasets. Overall, this paper makes a significant contribution to biologically-plausible deep learning.

**Questions:**

1. Your derivation showing the equivalence of the learning rule to backprop assumes the sigmoid scaling factor to be 4 (Line 5 of equations on top of Page 3 in Supplementary Section B). Is there any specific reason to do so? Could you please elaborate in the text. If not, could you kindly add that this is a specific assumption. Also, it would be nice to comment whether the scaling factor adds a bias to the learning rule when it's not 4.
2. What are the shaded lines in Fig 4. A & E? I initially thought they were stdev about mean or something. But if that's the case, why are they not centered about the mean line?
3. Could the Q weights being plastic, as opposed to Y, be motivated by the fact that inhibitory weights exhibit lower plasticity than excitatory weights? I understand that the plasticity can be instead imposed on Y but just wondering if there is a specific reason to induce plasticity in only one of them as opposed to both?
4. Doesn't having a full-rank Y matrix require the same number of interneurons as the number of pyramidal neurons in the layer? Is that biologically realistic?


**Limitations:**

The authors don't specifically mention any potential negative societal impact of their work but I don't think there are any straightforward implications either.

**Strengths And Weaknesses:**

Strengths:
1. The paper is generally well-written and addresses a very important topic in bio-plausible deep learning.
2. The authors present convincing evidence from neurophysiology to demonstrate the biological plausibility of their model architecture.
3. Along with the performance on image classification, the authors demonstrate how their algorithm solves toy tasks, specifically the XOR problem where the single-phase analog of burstprop fails.
4. The alignment to the backpropagation updates has interesting trends, especially when the corresponding performance is taken into account. However, this probably needs more investigation and would come under the scope of future work to try and understand how the alignment could be improved and if doing so would yield better performance.
5. The proposed model demonstrates impressive performance on image classification tasks (although it is unclear how this would scale to residual networks on big datasets) and incorporates a great deal of biological realism to make experimental predictions, thus contributing to the understanding of credit assignment in the brain.

Weakness:
1. I felt Section 3.3.1 could be better presented. It's a bit tedious to read right now and slightly unclear without looking into the derivations in the Appendix. It might be helpful if the authors could summarize the derivation as well as clarify the steps in the derivation in appendix to make this section more readable.
2. It is slightly unclear why the algorithms achieve similar performance although there is a difference in the alignment angle for MNIST. Is the model able to achieve good performance despite the misalignment with the gradient due to dataset size? Or is it because of the size of the parameter space? This issue is slightly more pronounced for CIFAR-10 where burstprop achieves better alignment but worse performance. I am wondering if this has to do with the hyperparameters, specifically burstprop using a higher lr compared to BurstCCNs?
3. Although the authors motivated the Q and Y matrices as feedback projections from higher layer pyramidal neurons and dendrite-targeting interneurons (inhibitory populations?), it is not necessarily imposed that weights in Q should be positive and Y should be negative. This is an issue that the authors already note in their Discusions section. I have another query related to this point which I have mentioned in the Questions section of the review (see Question #4 below).

---

> ### Author Response · Authors · 2022-08-02
> **Response to Reviewer QqnL (1 of 2)**
>
> >**“The proposed model demonstrates impressive performance on image classification tasks (although it is unclear how this would scale to residual networks on big datasets) and incorporates a great deal of biological realism to make experimental predictions, thus contributing to the understanding of credit assignment in the brain.”**
>
> This is an interesting suggestion by the reviewer and is something we are planning to explore. The BurstCCN can naturally be extended to include residual/skip feedforward connections if corresponding Q and Y feedback connections are also added to allow for backpropagation of errors. We have not yet experimented with such network structures but given the alignment with backprop we obtain in some conditions (see Fig. 6C) we believe that this would allow the BurstCCN to scale to more complex architectures and tasks.
>
> >**“I felt Section 3.3.1 could be better presented. It's a bit tedious to read right now and slightly unclear without looking into the derivations in the Appendix. It might be helpful if the authors could summarize the derivation as well as clarify the steps in the derivation in appendix to make this section more readable.”**
>
> We thank the reviewer for the suggestion, we have amended the text (see new Section 3.3.1) and clarified the steps of the derivation in the appendix (see SM, Section B).
>
> >**“It is slightly unclear why the algorithms achieve similar performance although there is a difference in the alignment angle for MNIST. Is the model able to achieve good performance despite the misalignment with the gradient due to dataset size? Or is it because of the size of the parameter space? This issue is slightly more pronounced for CIFAR-10 where burstprop achieves better alignment but worse performance. I am wondering if this has to do with the hyperparameters, specifically burstprop using a higher lr compared to BurstCCNs?”**
>
> There are a number of factors that could explain why the different models are able to achieve similar performance with differences in their alignment to backprop. We have some possible explanations for this result:
> 1. Our plots show a moving average of the alignment angle for each model. In some cases, this means that for some updates the alignment can be better for the BurstCCN than for Burstprop and vice versa even if one is less aligned on average. It is possible that these more closely aligned updates are larger or more important than those less closely aligned. The angle does not take into account the size of the updates.
> 2. The backprop gradient gives the direction of steepest descent. Both BurstCCN and Burstprop have updates that are consistently within 90 degrees of this direction which means that they are both making updates that descend the loss surface and will eventually converge to some local minimum. A lower alignment to backprop may be an indication that this convergence will be slower but, importantly, it does not necessarily suggest that the local minimum found will correspond to a higher error. BurstCCN and Burstprop follow different trajectories through the weight space which is not explained solely by the alignment angles. Despite this, we would like to emphasise the point that it is still important to look at the alignment angle as a measure of how well each model can backpropagate errors and as an indication of the expected performance.
> 3. The hyperparameters were selected independently for all models using Bayesian Optimisation (as mentioned in Appendix Section C.3). As the reviewer suggests, this could contribute to the mismatch between backprop alignment and performance.
>
> These are points that we will add to the discussion in the camera-ready version if space allows or to the SM otherwise. However, it is not clear to us why the dataset and parameter space sizes, as suggested by the reviewer, would contribute to this effect as they are the same across all models.

---

> ### Author Response · Authors · 2022-08-02
> **Response to Reviewer QqnL (2 of 2)**
>
> >**“Although the authors motivated the Q and Y matrices as feedback projections from higher layer pyramidal neurons and dendrite-targeting interneurons (inhibitory populations?), it is not necessarily imposed that weights in Q should be positive and Y should be negative. This is an issue that the authors already note in their Discussions section.”**
>
> Although our current work does not satisfy Dale’s law, we have a proposal of how this can be achieved through different cell types. This can be implemented with excitatory PC-to-PC connections and a separate feedforward inhibitory pathway composed of parvalbumin interneurons. These perisomatic-targeting interneurons are known to have STD dynamics which is required for the feedforward pathways of our model (Kinnischtzke et al.,2014). The STD feedback pathway can be implemented with excitatory PC-to-PC connections and a separate feedback inhibitory pathway composed of dendritic-targeting NDNF interneurons which display STD dynamics (Abs et al., 2018). The positive weight of the STF feedback pathway can be achieved through a VIP-SST-PC disinhibitory pathway and the negative weight simply through a feedback dendritic-targeting SST interneuron population. There is evidence for STF dynamics onto SST from both PC and VIP interneuron cell types (Kinnischtzke et al., 2014; Karnani et al., 2016). This is one of the reasons for why we motivate the Y weights as dendrite-targeting interneurons. We have preliminary results explicitly modeling these separate pathways to satisfy Dale’s law which obtains good performance on MNIST (3.4% test error).
>
> >**“Q1. Your derivation showing the equivalence of the learning rule to backprop assumes the sigmoid scaling factor to be 4 (Line 5 of equations on top of Page 3 in Supplementary Section B). Is there any specific reason to do so? Could you please elaborate in the text. If not, could you kindly add that this is a specific assumption. Also, it would be nice to comment whether the scaling factor adds a bias to the learning rule when it's not 4.”**
>
> Yes, the reason for adding this scaling factor of 4 is so that the gradient of the sigmoid is 1 when the dendritic potential is 0 (i.e. u=0). This prevents an implicit scaling down of the error signal by a factor of 4 by each layer in the backward pass. When this is not included, the network is heavily biased to learn in the later layers since earlier layers will have their error signals significantly scaled down. We will add this point to the main text for the camera ready version.
>
> >**“Q2. What are the shaded lines in Fig 4. A & E? I initially thought they were stdev about mean or something. But if that's the case, why are they not centered about the mean line?”**
>
> The shaded regions represent the standard error but this is displayed incorrectly due to an error when exporting the figure. Unfortunately, we only noticed this after submission and have now fixed it (see new Fig. 4).
>
> >**“Q3. Could the Q weights being plastic, as opposed to Y, be motivated by the fact that inhibitory weights exhibit lower plasticity than excitatory weights? I understand that the plasticity can be instead imposed on Y but just wondering if there is a specific reason to induce plasticity in only one of them as opposed to both?”**
>
> In our model, it is indeed possible to have top-down plasticity either on Q or Y. In the main paper, we have plasticity on Q and this could be motivated as stated by the reviewer. However, we interpret the feedback Y pathway as going through a population of dendritic-targeting interneurons which have shown to demonstrate this type of plasticity, i.e. cancellation to maintain an E-I balance, in a large number of studies (e.g. Froemke et al. 2007, Vogels et al. 2011 Science, D’Amour et al. Neuron 2015). We show in the Supp. Figure S1 that having this plasticity on Y is equivalent to having it on Q. The reason that the plasticity can be applied on either Q or Y weight but not on both is because the plasticity rule we use aims to silence the apical compartments. If all connections use this rule then it is trivial for all synaptic connections to just learn a weight of 0 which would lead to no useful error signals propagating backwards. If a different plasticity rule is used for at least one of the connection types, such as those cited in our discussion (e.g. Kolen-Pollack algorithm for learning weight symmetry), it is likely possible to have both connections being plastic without learning leading to this degenerate solution.
>
> >**“Q4. Doesn't having a full-rank Y matrix require the same number of interneurons as the number of pyramidal neurons in the layer? Is that biologically realistic?”**
>
> In the BurstCCN, each computational unit/node represents an ensemble of many pyramidal cells. For a full-rank Y matrix there must be at least one interneuron for every one of these pyramidal ensembles but this does not require there to be an equal number of each cell type.

---

> ### Comment · Reviewer_QqnL · 2022-08-08
> **Response to authors**
>
> I would like to thank the authors for taking the time to respond to all of my comments.
>
> 1. That sounds great. I would be excited to see future studies looking into scaling up this interesting architecture to large-scale tasks and extending it to other model architectures.
>
> 2.  Thank you for incorporating this suggestion.
>
> 3. I read the authors' response and I would like to point out a few additional points. I also believe that the effect of misalignment with the gradient in a non-convex optimization setting is not very well understood.
>    - The misalignment for smaller steps hypothesis seems reasonable: did the authors test it?
>    - A lower alignment to the gradient does not just imply slower convergence to a minima, but it could also lead to convergence to a different minima, specifically a sharper local minima that could imply worse generalization.
>    - Thank you for stating the hyperparameter selection procedure. I probably missed it in my initial read
>    - By dataset size and parameter space size, I actually meant that maybe these factors have an effect on what level of gradient misalignment could still yield good performance. But thank you for addressing my concern and I believe that better understanding these factors would require an in-depth study of the optimization process of BurstCCN.
>
> 4. I appreciate the authors' discussion about the possible neurophysiological components of implementing their model. The description is quite helpful to understand the motivations behind the specific design choices that the authors make in proposing their model. I must admit that I am not an expert in neurophysiology, but the authors' proposal looks convincing to me. The authors may already know of this work but I thought I will add a pointer to it in case it is helpful for them: https://openreview.net/forum?id=eU776ZYxEpz
>
> 5. The point about the scaling factor in sigmoid sounds good. It would be nice if the authors could add a sentence about the same in their camera-ready version.
>
> 6. That's great. It's good to know that the issue is fixed.
>
> 7. I appreciate the authors' explanation of what would happen if both Q and Y were allowed to be plastic. If I understand correctly, what the authors described is an intuitive explanation or is it an empirical finding? Nevertheless, I find this discussion to be quite useful in understanding the complexities of learning in the proposed model.
>
> 8. That's an interesting perspective of what the units in the model correspond to in a real brain. I understand the authors have probably not looked at the representations that their model learns in the intermediate layers. I believe it would be helpful if the authors could mention this correspondence in their manuscript so that future studies are aware of the subtlety while comparing representations learned by BurstCCN to those observed in the brain.

---

> > ### Author Response · Authors · 2022-08-09
> > **Response to Reviewer QqnL**
> >
> > Thank you for all of your helpful comments.
> >
> > > The misalignment for smaller steps hypothesis seems reasonable: did the authors test it?
> >
> > We have not tested it but this is something we are currently looking into and will add to the camera-ready version if we have conclusive results.
> >
> > > The point about the scaling factor in sigmoid sounds good. It would be nice if the authors could add a sentence about the same in their camera-ready version.
> >
> > Thank you, we will clarify this in the main text for the camera-ready version.

---

### Official Review · Reviewer_ChLd · 2022-07-14

**Rating:** 8
**Confidence:** 4
**Soundness:** 4 excellent
**Presentation:** 3 good
**Contribution:** 4 excellent

**Summary:**

Backpropagation, while allowing ANNs to learn the statistical regularities of their inputs, is not a biologically plausible method for how biological networks learn.  Different methods have arisen recently to bridge the gap between local Hebbian rules that are biologically realistic but limited in performance, and non-local rules like backprop that are not biologically realizable but perform closer to brain-like levels (or better).  The authors add to this field, providing a learning method that utilizes different spiking patterns ("events" and "bursts") and electrotonically separated compartments (somas, basal and apical dendrites) that, in one phase, is able to approximate backprop and learn complex visual labeling tasks in a more biologically plausible manner.

**Questions:**



**Limitations:**

Yes

**Strengths And Weaknesses:**

**Originality:**
As far as I know, no previous, biologically plausible learning rule has allowed for an approximation to backprop that has only one phase of learning.

**Quality:**
The authors pose their questions clearly and provide believable initial answers to these questions

**Clarity:**
The question, methods, and results of the study are all quite clear.

**Significance:**
By allowing for dynamic learning (see the sinusoid example) and biologically realistic modleing assumptions, the present work  of some of the most biologically realistic ANNs  The fact that they have been demonstrated to be able to learn in multiple layers in a single-phase makes a compelling case that the authors' setup may provide insight into a thorny biological problem

---

> ### Author Response · Authors · 2022-08-01
> **Response to Reviewer ChLd**
>
> We thank the reviewer for the encouraging feedback.

---

### Meta-Review · Area_Chair_oJsJ · 2022-08-23

**Recommendation:** Accept
**Confidence:** Certain

**Metareview:**

This paper describes a biologically plausible model of credit assignment in neocortical microcircuits, building on previous work in this area that uses apical dendrites and burst multiplexing. This model expands on these ideas and incorporates additional biological information related to short-term plasticity and cell-types to develop a model that can learn in a single phase, i.e. with no signal required to gate plasticity.

The reviewers were very positive about this paper and agreed that it makes a novel, insightful, and important contribution to the biological credit assignment field, and. As such, an accept decision was unanimously reached.

**Award:**

Yes

---

### Decision · Program_Chairs · 2022-09-14

Accept